# Molecular-level insights into the supramolecular gelation mechanism of urea derivative

**Shinya Kimura** [1] ✉, **Kurea Adachi**[1], **Yoshiki Ishii**[2], **Tomoki Komiyama**[1,3], **Takuho Saito** [4], **Naofumi Nakayama** [5], **Masashi Yokoya**[1], **Hikaru Takaya**[6,7], **Shiki Yagai** [8,9] ✉, **Shinnosuke Kawai** [3] ✉, **Takayuki Uchihashi** [2,10] ✉ & **Masamichi Yamanaka**[1,11]

Despite being a promising soft material embodied by molecular self-assembly, the formation mechanism of supramolecular gels remains challenging to fully understand. Here we provide molecular to nanoscopic insights into the formation mechanism of gel-forming fibers from a urea derivative. High-speed atomic force microscopy of the urea derivative revealed the presence of a lag phase prior to the formation of supramolecular fibers, suggesting a nucleation process. The fiber growth kinetics differ at both termini of the fiber, indicating a directional hydrogen-bonding motif by the urea units, which is supported by single-crystal X-ray crystallography of a reference compound. Moreover, we observed an intermittent growth pattern of the fibers with repeated elongation and pause phases. This unique behavior can be simulated by a theoretical *block-stacking* model. A statistical analysis of the concentration-dependent lag time on macroscopic observation of the gelation suggests the presence of a tetrameric or octameric nucleus of the urea molecules.

Gels are quintessential substances woven into the fabric of our lives, spanning a vast array of applications from daily necessities to foodstuffs[1]. Gels of highly functional polymers—those exhibiting remarkable mechanical strength, self-healing property, and dynamic modulation—have been realized through the meticulous design of their constituent polymers. The phenomenon of gelation often ensues from the self-assembly of organic compounds known as low-molecular-weight gelators (LMWGs) in an appropriate medium[2–5]. These gels, termed supramolecular gels, are constructed based on non-covalent interactions, and accordingly exhibit excellent

flexibility and stimuli responsiveness[3,6–8]. They are poised for applications in a variety of fields, encompassing pharmaceuticals[9–12], optoelectronic materials[13–15], and environmental purifications[16]. Within the three-dimensional network of a supramolecular gel, solvent molecules are entrapped and immobilized, and it has been believed that the process toward this network structure presumably proceeds in the following three steps (Fig. 1a): (I) LMWGs self-assemble one-dimensionally to form so-called supramolecular polymers; (II) these supramolecular polymers hierarchically assemble (bundle) to form mesoscopic fibers; (III) these fibers intertwine to

[1]Meiji Pharmaceutical University, Kiyose, Tokyo, Japan. [2]Department of Physics and Institute for Glyco-core Research (iGCORE), Nagoya University, Nagoya, Japan. [3]Department of Chemistry, Faculty of Science, Shizuoka University, Shizuoka, Japan. [4]Division of Advanced Science and Engineering, Graduate School of Science and Engineering, Chiba University, Chiba, Japan. [5]CONFLEX Corporation, Minato-ku, Tokyo, Japan. [6]Department of Life Science, Faculty of Life & Environmental Sciences, Teikyo University of Science, Adachi-ku, Tokyo, Japan. [7]Division of Advanced Molecular Science, Institute for Molecular Science, National Institute of Natural Science, Okazaki, Aichi, Japan. [8]Department of Applied Chemistry and Biotechnology, Graduate School of Engineering, Chiba University, Chiba, Japan. [9]Institute for Advanced Academic Research (IAAR), Chiba University, Chiba, Japan. [10]Exploratory Research Center on Life and Living Systems (ExCELLS), National Institutes of Natural Sciences, Okazaki, Aichi, Japan. [11]Deceased: Masamichi Yamanaka. ✉e-mail: s-kimura@my-pharm.ac.jp; yagai@faculty.chiba-u.jp; sskawai@shizuoka.ac.jp; uchihast@d.phys.nagoya-u.ac.jp

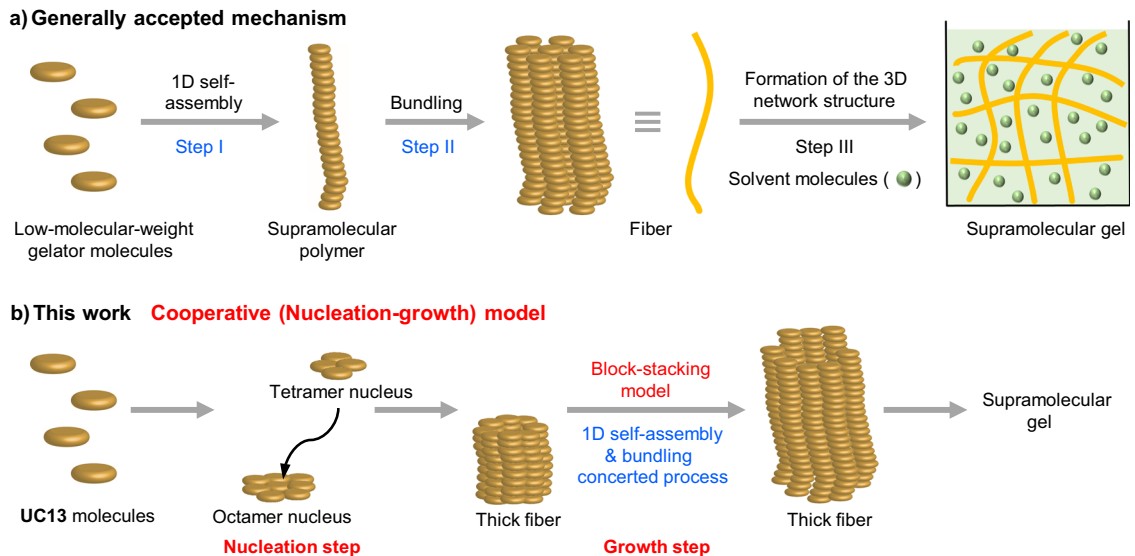

**Fig. 1 | Supramolecular gel formation. a** Generally accepted mechanism. **b** Mechanism supported by this work.

form a three-dimensional network with solvent molecules immobilized inside the network[17].

Although the above mechanism is plausible, it has been inferred from fragmentary results, and the whole aspect is still elusive. As pioneering advancements in properties and functionalities of covalent polymer gels have been made through understanding their structures, a profound grasp of the structure and formation mechanism of supramolecular gels is also paramount for their progressive innovation. The formation mechanism of supramolecular polymers has been rapidly elucidated by spectroscopic analysis and microscopic observation[18–23]. However, it is not yet clear whether similar mechanisms are applicable to the early stages of the formation of supramolecular gels. In particular, it has not been investigated whether mesoscopic fibers that form gels are formed directly or organized hierarchically from molecular-level aggregates[24–30], i.e., supramolecular polymers[31].

In the above context, the dynamic observation of the self-assembly of LMWGs would be important to completely understand the formation mechanism of supramolecular gels[32–34]. Hamachi and coworkers demonstrated that time-lapse imaging using confocal laser microscopes can provide critical insight into supramolecular gel formation[35]. The elongation of fibrous aggregates through a cooperative model was visualized for a dye-introduced LMWG. As another technique, high-speed atomic force microscopy (HS-AFM) is amplifying its significance as a powerful visualization tool for dynamic processes in nano-to-mesoscale with high spatiotemporal resolution. HS-AFM excels as a microscopy technique uniquely capable of visualizing dynamic behaviors of molecules in aqueous media even without labelling molecules[36,37]. The capabilities of HS-AFM in observing the functional dynamics of proteins have been well demonstrated, significantly advancing our comprehension of molecular processes[38,39]. Of late, HS-AFM has been harnessed to meticulously monitor the dynamism of artificial molecules, including supramolecular polymers[40–43] and metal-organic frameworks[44,45], thereby gleaning pivotal insights into their assembly processes. Accordingly, observing the processes of supramolecular gel formation using HS-AFM provides important insights into the formation mechanism of supramolecular gels.

In this study, we unveil the gel formation mechanism of *N*-tridecyl-*N'*-(2-benzylphenyl)urea (**UC13**)[46], LMWG[47–49], in dimethylsulfoxide (DMSO) and 1-ethyl-3-methylimidazolium bis(trifluoromethanesulfonyl)imide (EMI-Tf₂N), an ionic liquid, using HS-AFM (Figs. 1b and 2a). Although only substrate-guided self-assembly was observed in DMSO, we successfully captured the dynamic formation process of mesoscopic fibers following substrate-guided self-assembly in EMI-Tf₂N. Our HS-AFM imaging revealed intermittent growth with repeated elongation and pause phases, and this behavior was explained based on a *block-stacking* model devised in this study. We also found a strong concentration- and experiment-specific dependence of lag time prior to supramolecular gel formation, which is a diagnostic of cooperative (nucleation-growth) self-assembly process[21] initiated by a nucleation event. We thus attempted to estimate the nucleus size of the gelator molecules based on a statistical analysis of the concentration dependence of lag time, and revealed the two nucleus sizes corresponding to fiber formation and fiber growth processes.

## Results
### Gelation and HS-AFM observation of UC13
Previously, we showed that *N*-alkyl-*N'*-(2-benzylphenyl)urea derivatives (**UCn**, $n = 2–18$) are efficient LMWGs for a variety of organic solvents, revealing that the gelation ability is noticeably affected by the length of the alkyl chain (Fig. 2a)[46]. **UC12** and **UC13**, having dodecyl and tridecyl chains, respectively, formed gels in diverse organic solvents ranging from nonpolar *n*-hexane to polar DMSO. **UC13** in DMSO ($c = 50$ mM) afforded an opaque supramolecular gel within 10 min upon the natural cooling of a hot solution (hotplate temperature: 150 °C) to room temperature. However, as the concentration of **UC13** decreased, the time required for supramolecular gel formation increased. It took several days to a month to form a gel from a $c = 30$ mM solution in DMSO (*vide infra*, Fig. 2b and Supplementary Table 1). The concentration-dependent lag time in supramolecular gel formation and low volatility of DMSO are suitable for microscopic observation of the formation process of gel-forming supramolecular fibers by HS-AFM.

We initially attempted to visualize the dynamic self-assembly process of **UC13** in DMSO through HS-AFM. When an as-cooled 30 mM solution of **UC13** in DMSO was deposited on highly oriented pyrolytic graphite (HOPG) substrate, one-dimensional (1D) elongation of linear fibrils (height = 0.7 nm, widths = 7 nm) was visualized after several minutes (Fig. 2c, Supplementary Fig. 4, and Supplementary Movie 1). The height was consistent with the molecular thickness of **UC13**, whereas the widths were significantly longer than the longest molecular length of **UC13** (ca. 2.7 nm) (*vide infra*, Supplementary Fig. 5, and Supplementary Table 2). This finding indicates that these fibrils were composed of multistranded hydrogen-bonded chains of **UC13**. The assumption that the self-assembly of **UC13** is driven by hydrogen

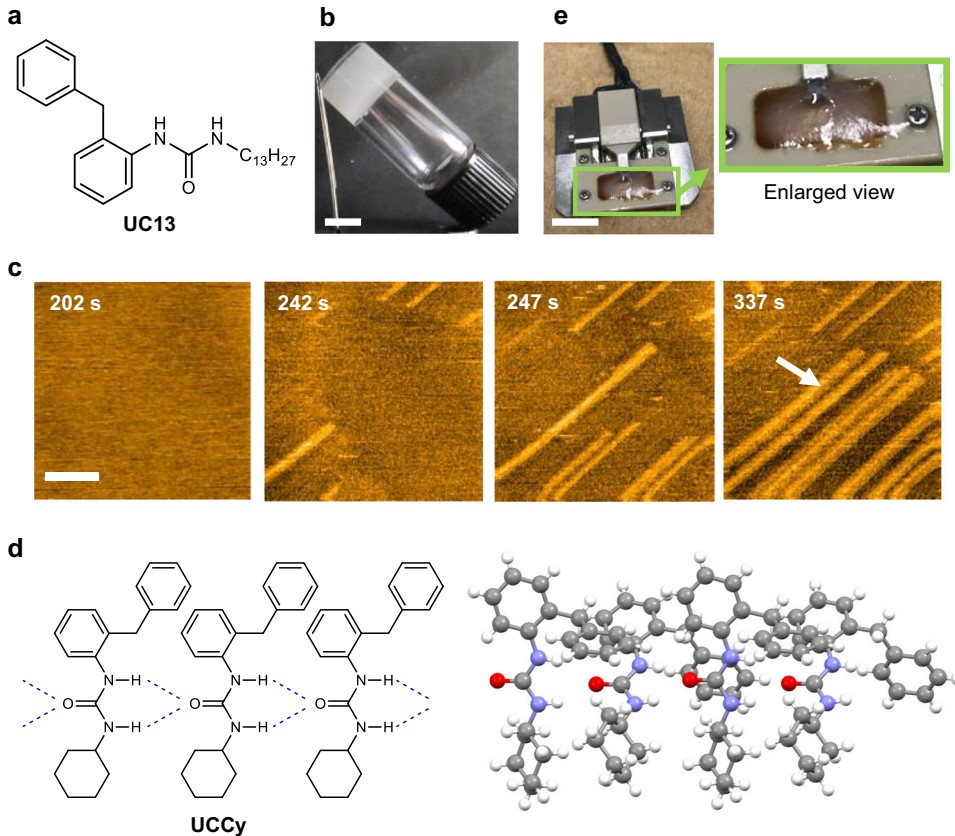

**Fig. 2 | HS-AFM observations of UC13 in DMSO. a** Structure of **UC13**. **b** Photograph of the supramolecular gel formed by 30 mM **UC13** in DMSO. Scale bar: 5 mm. **c** Clipped topographic HS-AFM images of 30 mM **UC13** in DMSO immobilized on a HOPG surface. The time indicated in each image shows the time elapsed since the addition of the **UC13** to the DMSO solution. Scale bar: 50 nm. Imaging rate: 0.5 s/frame. The latter stage fibril showing fast elongation, supported by already existing fibrils, is marked by a white arrow on the image at 337 s. See also Supplementary Movie 1. Representative time-lapse snapshots showing similar results from more than five independent experiments are shown. **d** X-ray crystallographic analysis of **UCCy**, the cyclohexyl analog of **UC13**. **e** Photograph acquired after measurements indicating that the supramolecular gel covers the substrate and cantilever. Scale bar: 10 mm.

bonding of ureido groups is based on single-crystal X-ray crystallography of the cyclohexyl analog, **UCCy** (Fig. 2d and Supplementary Fig. 6, See also Supplementary Fig. 7 for information on the X-ray scattering patterns of **UC13**). **UCCy** showed a 1D repeating structure consisting of a bimolecular unit with 1D orientation[50–53]. The distance between the N–H and C=O moieties of the ureido groups of each **UCCy** (ca. 3.1 Å) indicated that the intermolecular hydrogen bonding of these groups was the major driving force for self-assembly. The elongation of the fibrils composed of multiple chains of **UC13** was obviously supported by the crystal lattice of the HOPG substrate and proceeded at a rate of approximately 13 nm s$^{-1}$ in one direction. Assuming that the fibril elongation is driven by hydrogen bonding between urea units, this rate corresponds to an aggregation rate of ca. 30 molecules per strand per second. Interestingly, the elongation rate of the fibrils emerging in a latter stage (e.g., the fibril marked by the white arrow in the 337-s image in Fig. 2c), which could be supported by already existing fibrils, was markedly faster (> 30 nm s$^{-1}$), suggesting that secondary (lateral) interactions between fibrils may promote fibril formation[54–58]. At the end of the HS-AFM observation (after 40 min), gelation of the specimen solution was observed (Fig. 2e).

Because the minimum gelation concentration of **UC13** in DMSO is very high (30–50 mM), gelation occurs more instantly, making it difficult to capture the dynamic formation process of supramolecular fibers at the early stage of gelation. As an alternative low volatility solvent suitable for HS-AFM observation, we used an ionic liquid (EMI-Tf$_2$N) that could be gelled at much lower concentrations (1.8–5 mM). **UC13** in EMI-Tf$_2$N at $c = 2.0$ mM showed a sufficient lag time for

gelation, after which three distinct steps of self-assembly were observed (Fig. 3 and Supplementary Movie 2). In the first step, the two-dimensional (2D) growth of molecular sheets was observed on an HOPG substrate several minutes after adding the gelator into the solution (Fig. 3a). The second step involved the growth of short fibrils with a triangular pattern on the sheet for the following several minutes (Fig. 3b). After 1 h, the elongation of mesoscopic fibers with widths of approximately 20 nm was observed as the third step (Fig. 3c). These three steps are discussed in detail below. In this study, fibrous aggregates with sufficient thickness to constitute a gel are described as 'fibers', and the thinner fibrous aggregates are described as 'fibrils'.

## HS-AFM imaging of substrate-templated fibrils

Detailed AFM analysis revealed that the sheet-like structure that grew in the first step has a lamellar pattern with an interlayer spacing of approximately 2.7 nm according to 2D fast Fourier transform analysis (Fig. 4a and Supplementary Movie 3). The spacing is consistent with the length of the **UC13** molecule (Fig. 4a, Supplementary Fig. 5, and Supplementary Table 2), suggesting that the sheet-like structure is formed from hydrogen-bonded **UC13** chains arranged in two dimensions. These 2D arrangements of the hydrogen-bonded chains were not randomly distributed on HOPG but formed anisotropically with a triangular pattern because of epitaxial adsorption based on the honeycomb-like arrangement of the carbon atoms of HOPG (Fig. 3a)[59]. The average growth rate of the sheet-like structure of **UC13** molecules was estimated as approximately 14 nm s$^{-1}$ from the kymograph of sheet growth (Fig. 4b).

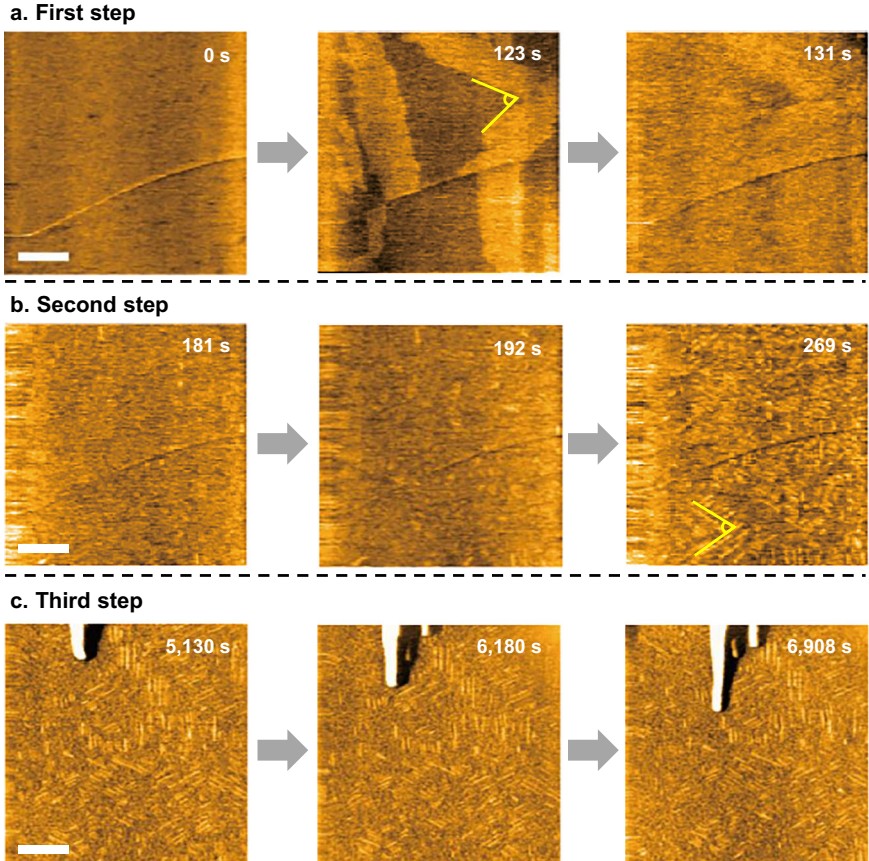

**Fig. 3 | HS-AFM observation of UC13 in EMI-Tf₂N. a–c** Clipped phase-contrast HS-AFM images of the fibrillation process of the 2.0 mM **UC13** in EMI-Tf₂N on the HOPG surface: **a** First step: the 2D growth of a unimolecular-thickness sheet; **b** Second step: the growth of short fibrils and the complete coverage of the substrate; **c** Third step: the elongation of thick bundled fibers. Scale bars: 100 nm. Imaging rate: (**a**, **b**) 1 s/frame and (**c**) 2 s/frame. See also Supplementary Movie 2. Representative time-lapse snapshots showing similar results from at least three independent experiments are presented.

In contrast to the first step, the growth of short fibrils in the second step (Fig. 4c and Supplementary Movie 4) was too fast to be imaged at the applied scanning speed (0.5 s/frame). This is most likely because the influence of the sheet-like structure on the substrate accelerated the self-assembly of **UC13**. The width of the short fibrils was estimated to be approximately 6 nm, based on the full width at half maximum of the cross-sections (Supplementary Fig. 8a, b). The fibrils grew on the substrate anisotropically, similar to the first layer in the first step, with the angle between the fibrils being approximately 60° (Supplementary Fig. 8c, d). Therefore, fibril growth was also indirectly affected by the substrate.

**HS-AFM of gel fiber: anisotropic growth and block-stacking model**
The third step involved the growth of fibers with a diameter of approximately 20 nm (Fig. 5a, Supplementary Fig. 9, and Supplementary Movie 5). The elongation rate of these fibers was 0.07 nm s⁻¹, considerably lower than that observed in the preceding steps (Fig. 5b). The fiber orientation was random and unaffected by the substrate, unlike that of the fibrils observed in previous steps (Fig. 5a and Supplementary Fig. 9a). Based on this observation and the concentration dependence described below, we believe the fibers observed in the third step were the elementary fibers causing physical gelation. The width of these fibers with tens of nanometers suggests that the gel-forming fibers are directly organized through multiple intermolecular interactions, not hierarchically via the bundling of hydrogen-bonded chains. Namely, the elongation of the fibers constituting the supra-molecular gel of **UC13** proceeds via a concerted mechanism featuring simultaneous vertical and horizontal growth.

In the third step, the growth of the thick fibers was significantly slower compared to the preceding steps. The time evolution of fiber growth is shown in Fig. 5b. The slope of the curve suggests an average growth rate of 0.07 nm s⁻¹. Interestingly, the kymograph of a single fiber (indicated by arrows in Fig. 5a) revealed that the growth rates of the two fiber ends were markedly different. Specifically, the average growth rate at the L-end was 0.06 nm s⁻¹, whereas it was only 0.01 nm s⁻¹ at the R-end (Fig. 5c and Supplementary Fig. 10). This finding indicates that the hydrogen-bonded urea chains of **UC13** are organized in a parallel manner to form fibers (Fig. 6a). This is because, in such supramolecular structures, the terminal functional groups of the fibers become C=O at one end and N–H at the opposite end. When considering the situation where free monomers bind to these fiber termini, the kinetic constant for this binding must depend on the rotation of the monomer around the two single bonds connecting the C=O and N–H groups (Fig. 6b). The C=O groups binding to the N–H end do not depend on their orientation and are therefore equal to the concentration of free monomers. In contrast, only structure B, where the two NH groups point in the same direction, can bind to the C=O end. Even if only the three structures in Fig. 6b are considered, the concentration of monomers capable of binding to the C=O end is statistically one-third of the free monomers. Consequently, the growth rate at the N–H end is statistically three times faster than that at the C=O end, which is consistent with the fact that the growth rates at the two fiber ends are unequal. On the other hand, if the rotational iso-merization of the monomer occurs sufficiently rapidly, the degree of conformational freedom at the fiber termini may be responsible. Specifically, the C=O terminus remains structurally invariant, whereas

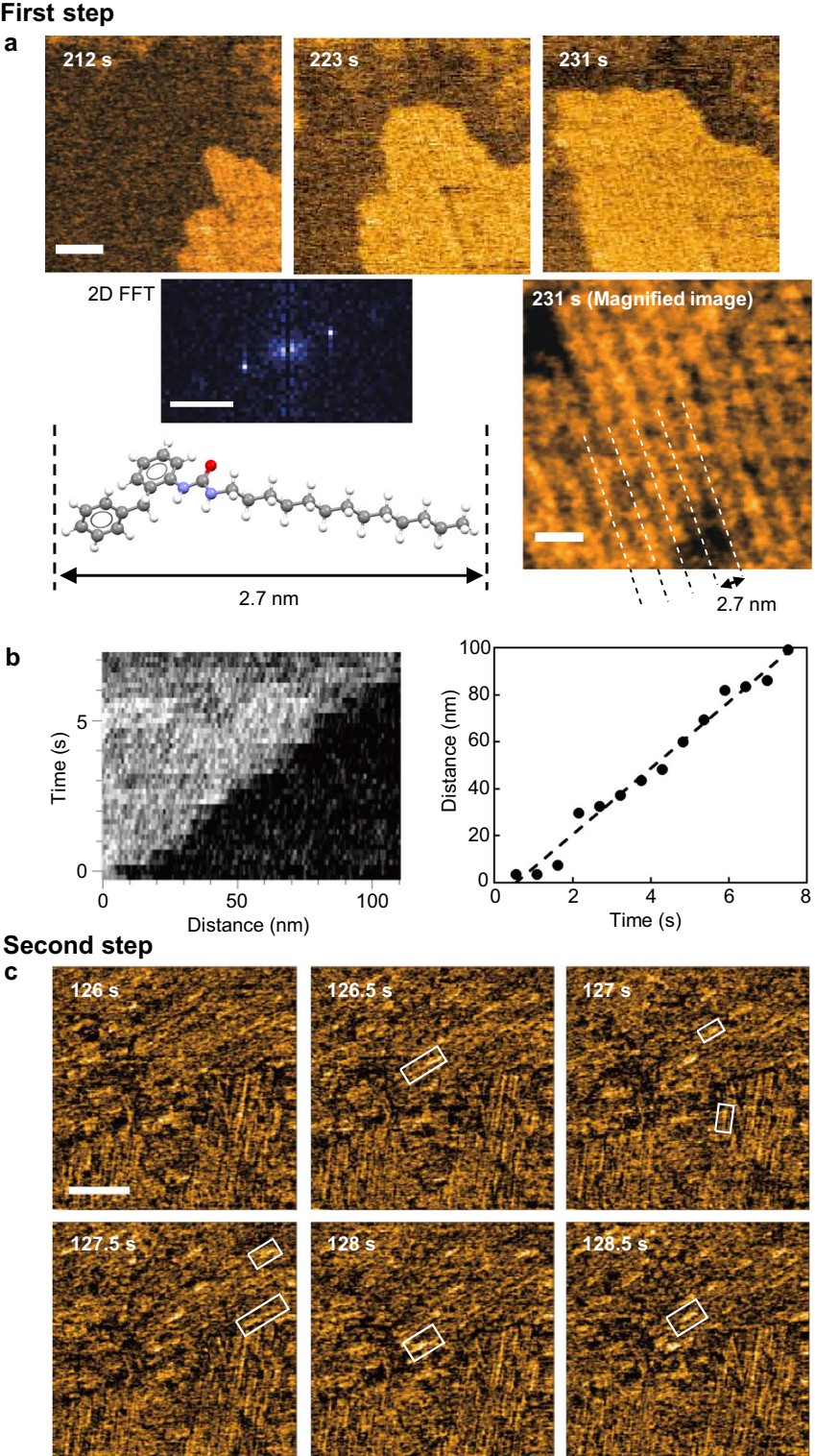

**Fig. 4 | First and second steps of the fibrillation process of UC13. a** Clipped phase-contrast HS-AFM images of the 2.0 mM **UC13** in EMI-Tf$_2$N in the first step. Scale bar: 20 nm. Imaging rate: 1 s/frame. Magnified phase-contrast image (scale bar: 5 nm). 2D fast Fourier transform (FFT) pattern (Scale bar: 0.5 nm$^{-1}$). Energy-minimized structure of **UC13**. See also Supplementary Movie 3. Representative snapshots from more than five independent experiments are shown. **b** Kymograph of the sheet-like structure formation in the first step and the growth distance of the sheet end as a function of time. Source data of the graphs are provided as a Source Data file. **c** Clipped phase-contrast HS-AFM images of the 2.5 mM **UC13** in EMI-Tf$_2$N in the second step. Fibrils appearing in the image have been enclosed in squares. Scale bar: 100 nm. Imaging rate: 0.5 s/frame. See also Supplementary Movie 4. Representative time-lapse snapshots showing similar results obtained from three independent experiments are presented.

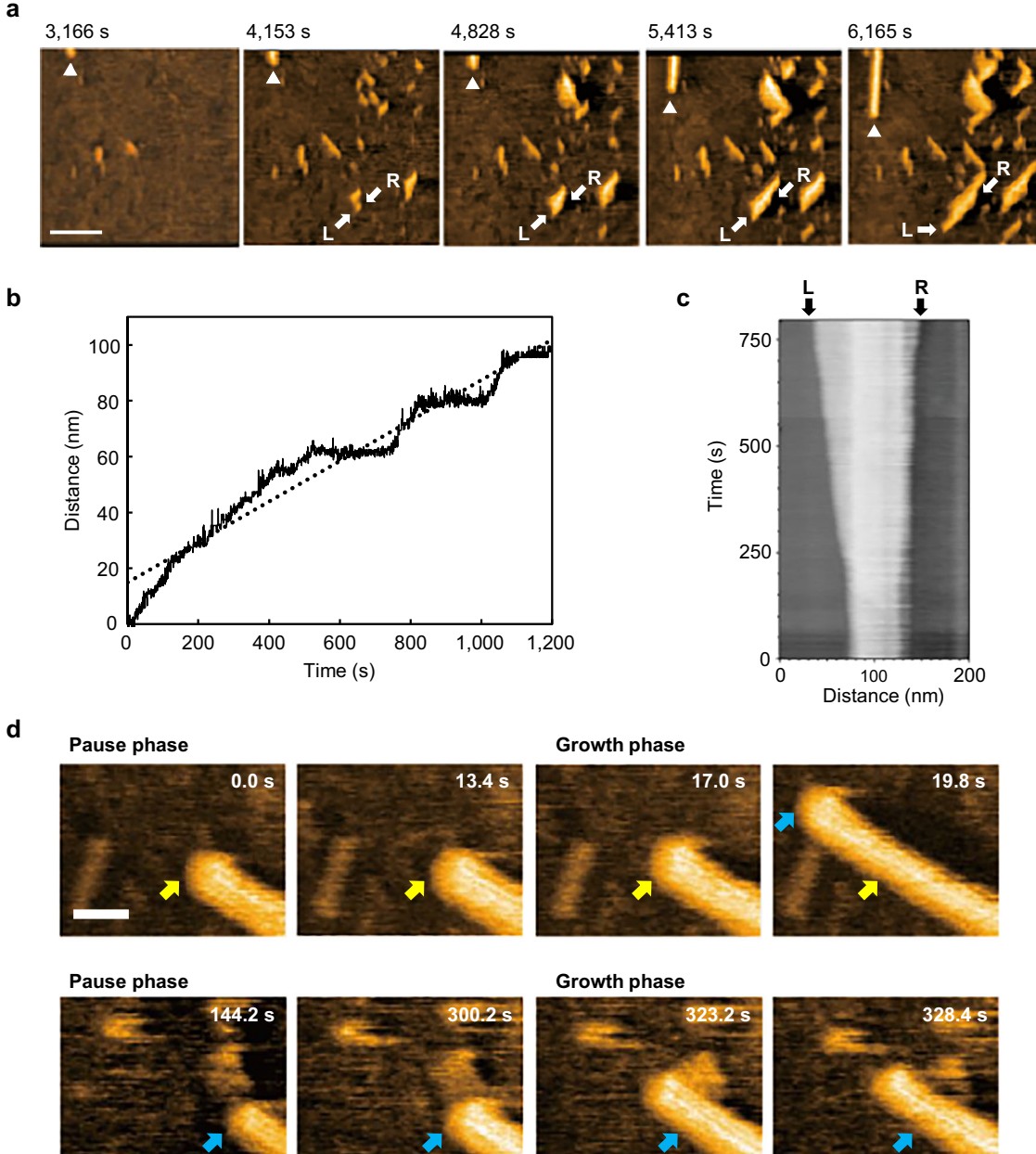

**Fig. 5 | Third step of the elongation process of UC13. a** Clipped phase-contrast HS-AFM images capturing the elongation of the thick fibers after the injection of 2.0 mM **UC13**. The time at which **UC13** was injected was defined as 0 s. Scale bar: 100 nm. Imaging rate: 0.5 s/frame. See also Supplementary Movie 5. Representative images from three independent experiments with similar results are shown.
**b** Elongation distance of a typical thick fiber, marked by the triangle in the image (**a**), as a function of the relative time. Here, the time of 0 s does not indicate the time elapsed since the molecule was injected. The average elongation rate, calculated from the linear fitting of the dashed line, was 0.07 nm/s. Source data of the graphs are provided as a Source Data file. **c** Kymograph showing growth at both ends of the thick fiber indicated by the arrow in the image (**a**). **d** Clipped phase-contrast HS-AFM images focusing on the fiber end in the 2.5 mM **UC13** in EMI-Tf$_2$N. Here, the time labels on the images indicate relative evolution time but do not mean the time elapsed since the molecule was injected. The yellow and blue arrows mark the positions of fiber ends during the pause phases prior to the onset of the growth phase. Scale bar: 30 nm. Imaging rate: 0.2 s/frame. See also Supplementary Movie 6. Representative images from time-lapse recordings of multiple independent experiments are shown.

the N–H terminus can undergo changes. In this case, contrary to the scenario described above, the growth rate at the C = O terminus could become faster. In either case, the rotational isomerization around the urea units is likely the origin of the anisotropic growth observed at the fiber termini.

Another interesting phenomenon observed in the third step is the discontinuous growth of fibers. HS-AFM images acquired at a rate of 0.2 s/frame revealed distinguishable growth and pause phases (Fig. 5d and Supplementary Movie 6). This might be related to the local consumption of monomers by fiber growth (growth phase) followed by the

homogenization of the monomer concentration via diffusion (pause phase). However, some pause phases were as long as 4 min (Fig. 5d, 144.2 s), which cannot be explained solely by the above mechanism.

Similar growth and pause phases have been reported for the formation of amyloid fibrils and discussed based on structural changes at the fiber ends[60,61] or the dock-lock model[62], both associated with the stabilization of attached protein monomers in the pause phases. Considering the present fiber is composed of much simpler **UC13** building blocks, we propose a *block-stacking* model (Fig. 7, Supplementary Methods) based on multidimensional intermolecular

**a**

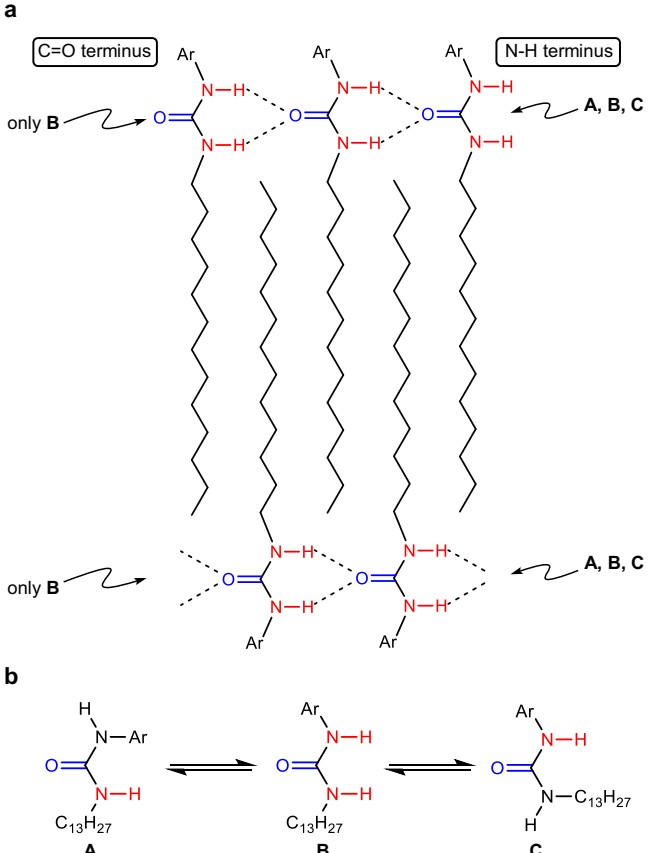

**Fig. 6 | Supramolecular structure and conformational isomerism of UC13.**
**a** Schematic representation of directional self-assembly of **UC13**. **b** Conformational equilibrium of **UC13**.

interactions from a more supramolecular viewpoint. The direct growth of 20 nm-wide fibers suggests that fiber growth was driven not only by urea-urea hydrogen bonding along the fiber axis, but also by van der Waals interactions between the alkyl chains or possibly CH-π interactions in the lateral (horizontal) direction (Supplementary Fig. 13).

In our proposed block-stacking model, the supramolecular fiber is modeled as a pile of "blocks," each of which represents a monomer (Fig. 7a). Growth of the fiber occurs by accommodating new monomers onto its ends from the solution phase. In the solution phase, gelators may exist not only as monomers but also as small aggregates (oligomers). For simplicity, in the present model we assume mixtures of monomers and dimers. If the fiber end facet is three-dimensionally fully packed by monomers, a newly bound monomer or dimer cannot benefit from the lateral interactions. This would result in reversible dissociation and binding, manifesting as the pause phase (Fig. 7a, step i). Once more oligomeric units benefiting from the lateral interactions attach to the fiber end facet like nucleation, it makes the facet irregular and gives rise to "reactive sites" (dotted squares in Fig. 7a, steps ii and iii) at which subsequent binding of monomers or dimers will be stabilized through the lateral interactions. This would manifest as the growth phase (Fig. 7a, step iii). The growth phase lasts until the fiber end facet is again stabilized by incidental full packing of monomers (Fig. 7a, step iv).

A numerical simulation was performed to verify this model using the following parameters: (I) monomers and dimers exist in the solution at a ratio of 90:10 and stochastically bind to the fiber ends, and (II) for the bound unit stabilized via horizontal interactions, the binding rate is 1000 times higher than that observed in the absence of such interactions. Supplementary Movie 7 shows the fiber growth

**a**

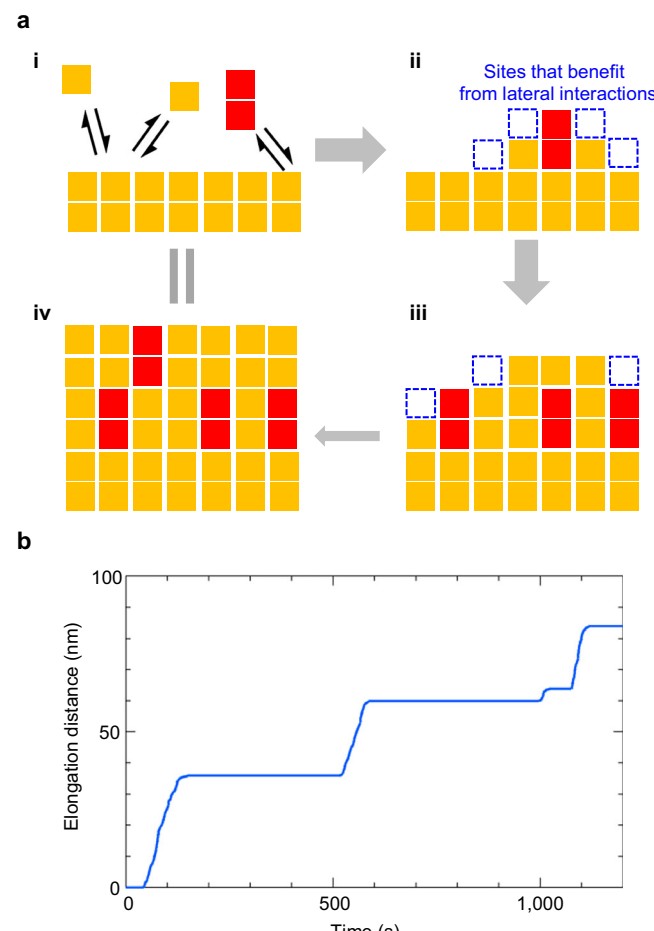

**b**

**Fig. 7 | Block-stacking model. a** Schematic illustrations of the block-stacking model to explain the growth of fibers in the third step. **b** Growth distance as a function of time obtained by numerical simulation of the block-stacking model. See also Supplementary Movie 7. Source data of the graphs are provided as a Source Data file.

simulated using this model, and Fig. 7b presents a plot of the elongation distance against time. The simulated fiber growth reproduces the experimentally observed fiber growth shown in Fig. 5b well in terms of the alternating emergence of the elongation and pause phases.

## Nucleation process of UC13

So far we have investigated the fiber growth process microscopically. Here, we investigate how this fiber formation is initiated through not only microscopic but also macroscopic observations. For this purpose, we initially studied how microscopic **UC13** aggregates emerge in EMI-Tf$_2$N ($c = 3.0$ mM) through time-dependent dynamic light scattering (DLS) measurements (Supplementary Fig. 14a, b). Upon cooling the hot solution to 25 °C, only small particles with hydrodynamic diameters around 7 nm were detected until 2100 s (35 min), which can be attributed to the monomers or small oligomers of **UC13**. After this lag phase, aggregates with hydrodynamic diameters above 100 nm suddenly appeared at 2400 s (40 min). This is a typical observation for cooperative (nucleation-growth) self-assembly mechanism of molecules[18]. In line with this result, the storage moduli ($G'$) and loss moduli ($G''$) of the EMI-Tf$_2$N solution of **UC13** ($c = 2.5$ mM), determined by dynamic viscoelasticity measurements using a rheometer[63], exhibited sigmoidal increases after 3600 s (60 min) (Supplementary Fig. 14c). The $G'$ value exceeded 1.0 kPa at 6000 s (100 min) and then saturated with 1.9 kPa at 8400 s (140 min).

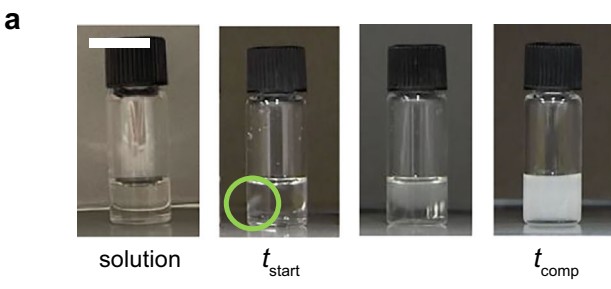

**a**

solution $t_{start}$ $t_{comp}$

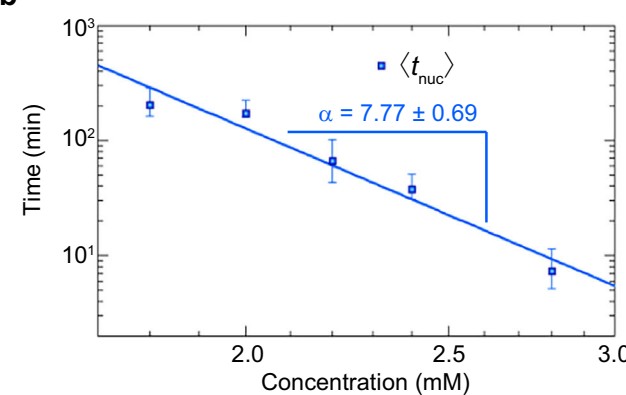

**b**

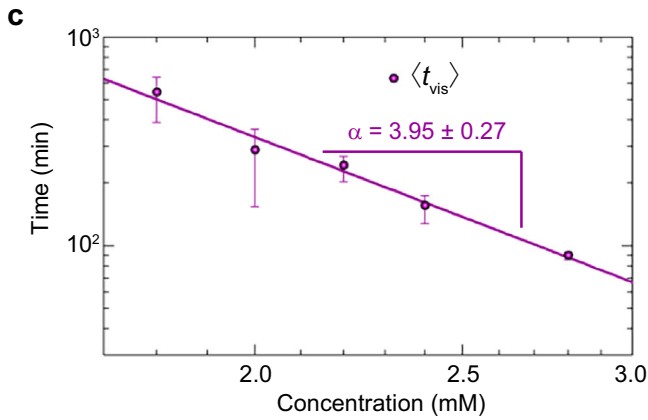

**c**

**Fig. 8 | Relationship between gelation time and concentration of UC13.**
**a** Definition of $t_{start}$ and $t_{comp}$. Scale bar: 10 mm. See also Supplementary Movie 8.
**b** Log-log plot of the nucleation time vs. concentration obtained from macroscopic observation of gel formation. Fitted line to the power law $\langle t_{nuc} \rangle \propto [\mathbf{UC13}]^{-\alpha}$, which becomes a straight line $\log \langle t_{nuc} \rangle = -\alpha \log[\mathbf{UC13}] + \text{const.}$ in the log-log plot, is shown with the value of the exponent $\alpha$. Error bars show 95% confidence intervals estimated by the bootstrap method. The uncertainties in the fitted exponents are the 1σ error estimated by the bootstrap method. Source data of the graphs are provided as a Source Data file. **c** Log-log plot of the time required for the gel to grow into a visible size vs. concentration obtained from the observation of macroscopic gel formation. Fitted line to the power law $\langle t_{nuc} \rangle \propto [\mathbf{UC13}]^{-\alpha}$, which becomes a straight line $\log \langle t_{nuc} \rangle = -\alpha \log[\mathbf{UC13}] + \text{const.}$ in the log-log plot, is shown with the value of the exponent $\alpha$. Error bars show 95% confidence intervals estimated by the bootstrap method. The uncertainties in the fitted exponents are the 1σ error estimated by the bootstrap method. Source data of the graphs are provided as a Source Data file.

The presence of nucleation process was further confirmed by a seeding experiment for the lag phase of **UC13** in EMI-Tf$_2$N. When a tiny amount of a separately prepared gel was added to the lag phase solution ($c = 1.8$ mM), the gelation duration decreased from 19 to 10 h (Supplementary Fig. 15)[58].

In a solution without aggregates, molecules exist in a super-saturated state. When the formation of nuclei, which serves as

templates for the subsequent growing process, occurs, aggregate formation propagates throughout the system. To shed light on the nucleation process, we studied the concentration dependence of the lag time in the supramolecular gel formation of **UC13** in EMI-Tf$_2$N. A closer inspection of the gelation process in a vial by movie revealed that the turbidity caused by the aggregation of **UC13** occurred at a single point in the homogeneous solution, and gelation was completed when the turbidity had spread throughout the system (Fig. 8a and Supplementary Movie 8). Therefore, we defined the gelation start time ($t_{start}$) as the time when the turbidity occurred visibly, and the gelation completion time ($t_{comp}$) as the time when the turbidity had spread throughout the system. Details of the algorithm used to determine $t_{start}$ and $t_{comp}$ from digitally recorded movies are given in Supplementary Methods. Statistics of ten samples were taken for each concentration and the average and standard deviation of $t_{start}$ and $t_{comp}$ were obtained (Supplementary Table 3). For the 2.2 mM solution, $t_{start}$ and $t_{comp}$ were $311 \pm 67$ min and $604 \pm 117$ min, respectively, which became shorter ($t_{start} = 97 \pm 8$ min; $t_{comp} = 155 \pm 8$ min) upon a concentration increase to 2.8 mM. In contrast, $t_{start}$ and $t_{comp}$ became longer to $750 \pm 205$ min and $1077 \pm 248$ min, respectively, upon a concentration decrease to 1.8 mM. At concentrations below 1.5 mM, no supramolecular gel formation was observed, and a suspension of fibrous macroscopic aggregates was obtained instead (Supplementary Fig. 16). At 1.0 mM, no macroscopic aggregate formation was observed, and the solution remained homogeneous.

Despite the carefully controlled experimental conditions, considerable experimental variations in $t_{start}$ (also in $t_{comp}$) were observed (Supplementary Table 3). This variation was possibly related to the experimental observation that gelation started at a single point in each solution. This implies that the supramolecular gel formation is dominated by the formation of a single nucleus in the entire solution. Once a nucleus is formed, turbidity rapidly spreads throughout the system before another nucleus forms. The large variation in $t_{start}$ can be attributed to statistical fluctuations in nucleation because statistical averaging is ineffective if nucleation occurs only once in each sample.

As the nucleus is expected to be a microscopic object consisting of a small number of **UC13** molecules, unlike objects discernible by the naked eye, the observed $t_{start}$ can be modelled using the following equation:

$$t_{start} = t_{nuc} + t_{vis}, \tag{1}$$

where $t_{nuc}$ is the time for nucleation and $t_{vis}$ is the time required for the gel to grow from the nucleus into the smallest macroscopic size discernible by the naked eye. As discussed above, nucleation is subject to statistical fluctuation in bulk experiments, whereas the subsequent gel growth is a macroscopic process in which fluctuation would be negligible, as described by $\text{SD}[t_{vis}] \approx 0$, where SD[] denotes the standard deviation. Therefore, by considering the standard deviation of both sides in Eq. (1), we obtain:

$$\text{SD}[t_{start}] = \left( \text{SD}[t_{nuc}]^2 + \text{SD}[t_{vis}]^2 \right)^{1/2} \approx \text{SD}[t_{nuc}]. \tag{2}$$

Further, if $t_{nuc}$ follows an exponential distribution, which is a consequence of nucleation kinetics, as described in the Supplementary Methods, its standard deviation is related to the average as:

$$\text{SD}[t_{nuc}] = \langle t_{nuc} \rangle. \tag{3}$$

Combining Eqs. (2) and (3), we obtain:

$$\text{SD}[t_{start}] \approx \langle t_{nuc} \rangle. \tag{4}$$

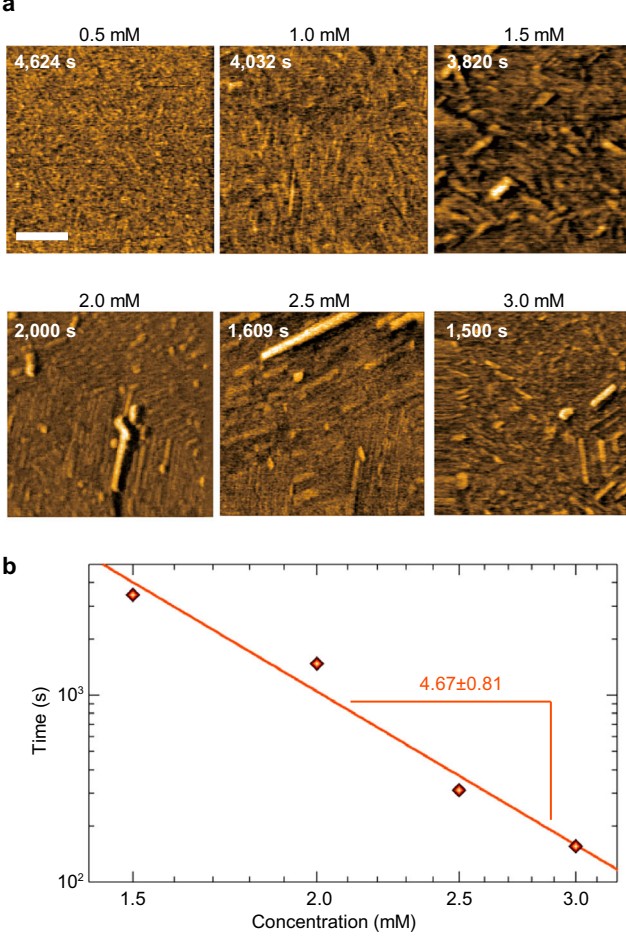

**a**

0.5 mM — 4,624 s
1.0 mM — 4,032 s
1.5 mM — 3,820 s
2.0 mM — 2,000 s
2.5 mM — 1,609 s
3.0 mM — 1,500 s

**b**

4.67±0.81

Time (s) — $10^3$, $10^2$

Concentration (mM) — 1.5, 2.0, 2.5, 3.0

**Fig. 9 | Relationship between the appearance time of third-step fibers and concentration of UC13. a** AFM images obtained at various **UC13** concentrations of 0.5, 1.0, 1.5, 2.0, 2.5, and 3.0 mM. Scale bar: 100 nm. Representative images from more than three independent experiments are shown. **b** Log-log plot of the first appearance time of third-step fibers in the AFM observation vs. initial monomer concentration in the solution. The uncertainty in the fitted exponent is the 1σ error estimated by the bootstrap method. Source data of the graphs are provided as a Source Data file.

Equation (4) implies that we can determine the average time for microscopic nucleation, $\langle t_{nuc} \rangle$, from the data on the macroscopically observed $t_{start}$ by examining its statistical fluctuation.

Figure 8b shows a plot of $\langle t_{nuc} \rangle$ obtained from the analysis described above, as well as the corresponding power-law fit $\langle t_{nuc} \rangle \propto [\text{UC13}]^{-\alpha}$. According to this figure, $\langle t_{nuc} \rangle$ has a power-law dependence on the monomer concentration, with an estimated exponent of 7.77 ± 0.69, where the uncertainty is the 1σ error estimated using the bootstrap method[64]. This exponent indicates a nucleus size of 8 (or 6–10, considering statistical uncertainty). While this octamer nucleus may exist as an isolated octamer in solution, as depicted in Fig. 1b, another possibility has been pointed out in the literature[65–67] regarding the interpretation of concentration dependence by nucleus size. The alternative suggests an octameric substructure embedded in a larger aggregate, with the nucleation corresponding to the formation of such substructures through conformational conversion inside the aggregate. This possibility becomes particularly relevant when interpreting non-integer exponents[67]. However, given the statistical uncertainty, the present estimate of 7.77 ± 0.69 for the exponent is not sufficiently statistically accurate to conclude a non-integer exponent. Having obtained $\langle t_{nuc} \rangle$, we also evaluated $t_{vis}$ by subtracting $\langle t_{nuc} \rangle$ from $\langle t_{start} \rangle$ in Eq. (1). Figure 8c

shows the plot of $\langle t_{vis} \rangle$ versus the monomer concentration. The time follows a power-law dependence with an exponent of 3.95 ± 0.27. These analyses indicate that aggregate formation involves (at least) two steps, one proceeding through octamer nuclei formation and another through tetramer nuclei formation.

We then applied a similar approach to the formation of the third-step thick fibers observed by HS-AFM (Fig. 5a). For this purpose, fibril and fiber formation by **UC13** was examined by HS-AFM in the range of 0.5–3.0 mM (Fig. 9a). For 0.5 mM and 1.0 mM solutions, only sheet-like structures and short fibrils corresponding to the first and second steps, respectively, were observed, but the thick fibers observed in the third step were not detected even after prolonged observation. The third-step thick fibers were observed for the 1.5–3.0 mM solutions. This result is consistent with that of macroscopic observations, in which supramolecular gel formation was not observed below 1.8 mM, and $t_{start}$ became shorter with increasing **UC13** concentration (Supplementary Fig. 17b). Thus, the thick fibers found in the third step were identified as the primary components of the supramolecular gels. Importantly, the elongation and pause phases were also observed, but their rate and duration were concentration-independent (Supplementary Fig. 18). In contrast, the appearance time of the thick fibers became shorter with increasing **UC13** concentration (Supplementary Fig. 17c). We thus conclude that the concentration dependence of fiber appearance reflects the time required for nucleation rather than fiber growth.

Figure 9b presents a log-scale plot of the appearance time of the thick fibers shown in Supplementary Fig. 17c versus concentration. The least-squares fitting to the power-law dependence yielded an exponent of 4.67 ± 0.81, which is close to that of $\langle t_{vis} \rangle$ rather than to that of $\langle t_{nuc} \rangle$.

As described above, we identified the specific exponents based on the observations of the concentration dependences of three different time scales: the start of fiber growth via nucleation ($t_{nuc}$), fiber growth into a visible size ($t_{vis}$), and the appearance of third-step thick fibers in the HS-AFM experiments. These values allowed us to consider two possible mechanisms. The first mechanism (Fig. 10, path a) assumes that gel formation is initiated by the formation of an octamer nucleus, followed by growth of the nucleus to form a short fiber. The resulting fiber may act as a template and recruit the nucleus with a smaller nucleus size of 4 (or 3–5)[19–23]. Fibers can also grow on the surface of probe or HOPG-substrate in HS-AFM experiments. These surface reactions may resemble those on existing fibers and proceed via the tetramer nuclei. Thus, the octamer nucleation step can be skipped under these conditions. This explains the fourth-power dependence of the fiber appearance time in HS-AFM images and a reason why the fibers appear faster than the macroscopically observed nucleation time. In addition, the existing fibers may cause not only the growth of fibers on their termini, but also the formation of new fibers on the surface through secondary interactions[54–58].

The second mechanism (Fig. 10, path b) is that some oligomers may have formed as precursors before gel formation. These precursor oligomers may form via tetramer nuclei, resulting in a fourth-power dependence of the precursor concentration on the initial monomer concentration. Two of these precursors may then associate with each other to form a nucleus for gel-fiber growth. The latter is the rate-limiting step for the entire process, explaining the eighth-power dependence observed for $\langle t_{nuc} \rangle$. Upon nucleation, the fiber grows via the accumulation of precursors on the existing fiber at a rate proportional to the precursor concentration, that is, the fourth power of the initial monomer concentration. By identifying the thick fibers observed in the third step of the HS-AFM results as the precursor oligomers conjectured here, this scenario can also explain the fourth-order concentration dependence and rapid appearance time of the third-step fibers.

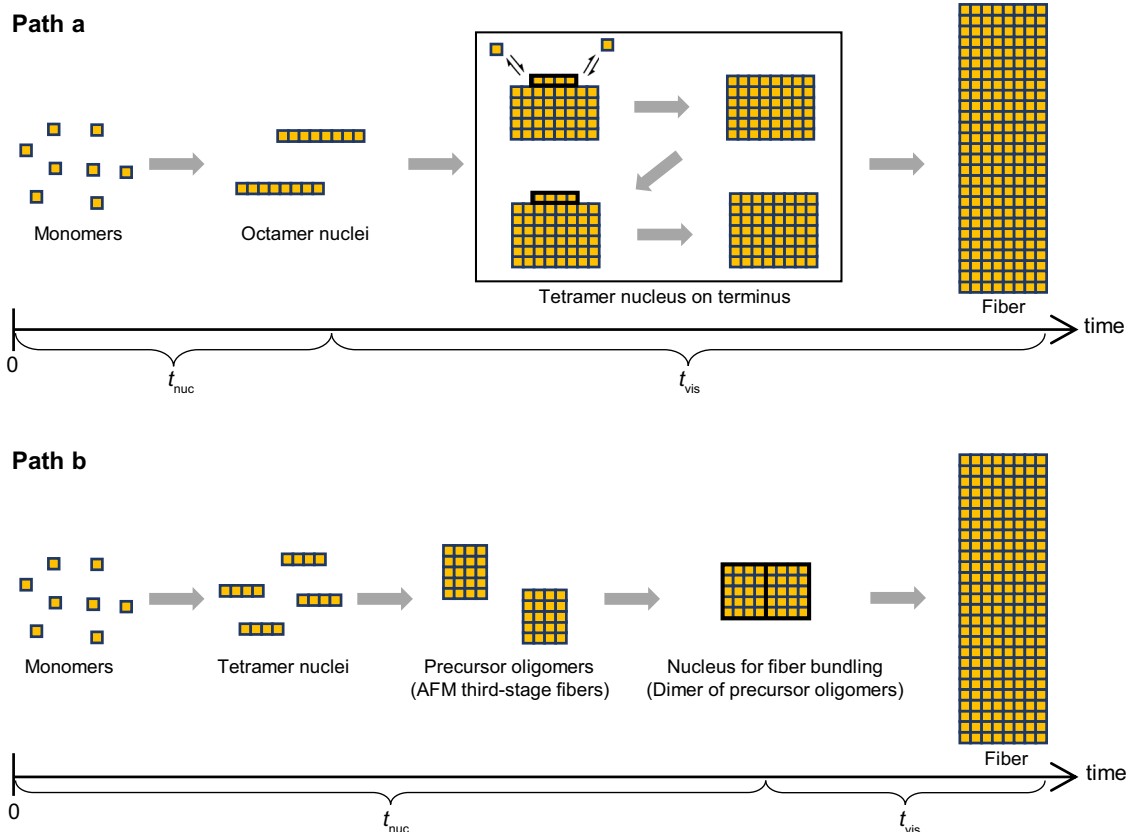

**Fig. 10 | Schematics of the two proposed mechanisms for gel formation. a** Pathway initiated by the formation of octameric nuclei. **b** Pathway initiated by the formation of tetrameric nuclei.

## Discussion

Although there have been many studies on the properties and applications of supramolecular gels, our understanding of their formation process remains limited. In this study, we succeeded in dynamically capturing the fiber formation process of a urea gelator through HS-AFM. We found that the organization of the gelator molecules by interaction with the substrate occurs before the formation of the gel-forming fiber. The subsequent substrate-independent growth of gel-forming fibers was found to be not a simple process in which both ends grow equally and continuously, but a discontinuous process in which the growth and pause phases are repeated in a directional manner. To rationally explain these observations, we discussed the dynamic change in molecular conformation based on the single-crystal structure of a reference urea compound, and also proposed a theoretical model called the block-stacking model. Furthermore, statistical analysis based on the macroscopic observation of gelation time as a function of concentration revealed that the urea gelator forms a supramolecular gel based on the cooperative (nucleation-growth) model.

The overall mechanism of supramolecular gel formation of **UC13** is as follows (Fig. 1b): **UC13** forms an octamer nucleus, which leads to the formation of bundled fibers by accumulating tetramer nuclei on the existing fiber. The fibers formed from these nuclei grow in an intermittent manner repeating the elongation and pause phases, which could be explained by our block-stacking model. These processes follow the cooperative mechanism[68–70]. To conclude, a similar approach to the present study could be applied to other versatile supramolecular gels, especially those that have already proven useful, allowing us to obtain insights into the molecular-level mechanism of their formation. This would lead to more spatiotemporal control of gel formation, which would greatly expand the range of gel applications.

## Methods

### General
Single-crystal X-ray crystallographic analyses were performed at SPring-8 beam line BL40XU with Si (111) monochromated synchrotron radiation (0.83136 Å) using DECTRIS EIGER X 1M detector. Rheological measurements were performed using TA Instruments DHR 2.

### Materials
**UC13** was synthesized according to our previous report (See Supplementary Methods)[46]. DMSO was purchased from NACALAI TESQUE, INC., and EMI-Tf$_2$N was purchased from Tokyo Chemical Industry Co., Ltd. (TCI).

### High-speed atomic force microscopy (HS-AFM)
HS-AFM experiments were conducted with a laboratory-built instrument, utilizing an Olympus BC-AC7 microcantilever in the tapping mode. The microcantilever, with a nominal spring constant of 0.2 N/m and a resonant frequency of approximately 600 kHz in solution, is not originally equipped with a sharp tip. HOPG was chosen as the observation substrate. In the HS-AFM experiments, we obtained a phase image as well as a topographic image. Phase images often provide better contrast and sharper images than topographic images in fiber imaging (Supplementary Fig. 1). Therefore, the discussion in this article has been based on phase images, unless stated otherwise.

### Data availability
Crystallographic data for **UCCy** have been deposited with the Cambridge Crystallographic Data Centre under the deposition number CCDC-2321499. The data can be obtained free of charge from the Cambridge Crystallographic Data Centre via www.ccdc.cam.ac.uk/data_request/cif. All data necessary to evaluate the conclusions of the paper are present in

the paper and/or the Supplementary Materials. Source Data are provided with this paper, and additional data may be requested from the corresponding authors. Source data are provided with this paper.

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

## Acknowledgements

This work was supported by Grant-in-Aid for Scientific Research (no. 23K14327 for S.Ki.; 17H06373 and 21K05105 for N.N.; 18H04512, 20H04669 for T.U.; 17H06374 and 21K06485 for M.Ya.) the Japan Society for the Promotion of Science (JSPS) or the Ministry of Education, Culture, Sports, Science and Technology (MEXT), the JSPS for a Research Fellowship for Young Scientists (no. 21J20988 for T.S.), and a CREST Grant-in-Aid (JPMJCR21L2) from the Japan Science and Technology Agency (JST) for T.U. This work was also supported by JSPS KAKENHI in a Grant-in-Aid for Transformative Research Areas "Materials Science of Meso-Hierarchy" (no. 24H01729 for S.Ki.; 23H04873 for S.Y.). Professor Tomohiro Seki in Shizuoka University is acknowledged for his insightful comment into the nucleation process which motivated the analyses of gelation times. Part of the computation was performed by the supercomputers of ACCMS, Kyoto University. We thank Ms. Yoko Harada (Shizuoka University) for her help with the preliminary experiments.

## Author contributions

S.Ki. and M.Ya. designed the project. T.K. performed the gelation experiments using DMSO. K.A. carried out the gelation experiments using EMI-Tf$_2$N. Y.I. and T.U. collected the HS-AFM data. T.U. wrote the HS-AFM section of the manuscript. N.N. simulated the conformation of **UC13** and its dimer. H.T. collected the XRD data. T.S. and S.Y. measured the DLS data. S.Ka. proposed and simulated the block-stacking model, and estimated nucleus sizes. S.Ka. wrote the simulation section of the manuscript. S.Ki., S.Y. and M.Ya. prepared the overall manuscript including figures. All authors including M.Yo. have contributed by commenting on the manuscript. The overall project was directed by S.Ki., S.Y. and M.Ya. We dedicate this paper to the memory of our co-author, Professor Yamanaka, who sadly passed away during the preparation of this manuscript. This publication serves as a testament to his lasting academic legacy.

## Competing interests

The authors declare no competing interests.
