## [Transparent Peer Review file · Nature Communications]

Molecular-Level Insights into the Supramolecular Gelation Mechanism of Urea Derivative

Corresponding Author: Professor Shinya KIMURA

Version 0:

Reviewer comments:

Reviewer #1

(Remarks to the Author)

The article by Kimura et al. presents molecular insights into the widely studied supramolecular gelation process using high-speed AFM imaging performed in the native liquid phase. This work chooses a known urea derivative to study the self-assembly and gelation and provide unique understanding into the transformation from nanoscopic to microscopic and macroscopic scale. They show asymmetrical growth kinetics of the two end of the fibers, and repeated elongation-pause phase in the fiber growth, three steps involved in the gelation and the substrate induced effects in the self-assembly. The authors finally propose a theoretical block-stacking model to understand the observation. Overall, the manuscript is one of the very few examples of supramolecular fiber formation probed with nanoscale details in solution and provides some useful information on the process of self-assembly of the chosen molecule. However, the conclusions drawn from the work is specifically relevant to this molecule and it cannot be generalized. Furthermore, the substrate effect seems to play some role and thus it may not be obvious to extrapolate all observations to the processes happening in the bulk. Also, some conclusions drawn are not fully convincing. The reviewer feels that the manuscript is not easy to read and understand because of the lack of some information (described below) at different places. Thus, I do not support its publication in Nature Communication, at least not in its current form.

1. Would the author be able to visualize any other class of gels, even the very simplest known ones, or their method is specific to this gel?
2. In fig. 1b, octameric nucleus seems to be made of 8 monomers, but in the manuscript authors repeatedly mention "It should be noted that this octamer nucleus is not necessarily an isolated octamer in the solution; it might be an octameric substructure within a larger oligomer."
3. "At the end of the HS-AFM observation (after 40 min), macroscopic supramolecular gel formation was observed..." This is confusing considering that in Table 1 we see the gelation time in weeks and here it is happening in 40 min.
4. "The low gelation ability of UC13 in DMSO (minimum gelation concentration: 30–50 mM) is not suitable for the HS-AFM observation of much mesoscopic fibers inducing physical gelation." Not clear why is it so.
5. " In the first step, the two-dimensional (2D) growth of unimolecular-thickness sheets was observed" How is this deciphered? Authors should consider providing the height color scale of this AFM image and the topographical height plot.
6. "second step involved the growth of short fibrils with a triangular pattern " Author should consider highlighting the triangular pattern on the image in Fig. 3b.
7. Can such imaging be performed on other substrate like Mica?
8. Authors suggest the possibility of π - π or possibly CH- π . Does it show any change in the UV-Vis absorption spectra?
9. " This implies that the supramolecular gel formation is dominated by the formation of a single nucleus in the entire solution." This is a very bold statement, and the supporting data is not very convincing. Firstly, this reviewer believes that manual and visual probing of the t_{start} and t_{vis} is prone to a lot of errors as is also evident. Its difficult to manually point out at what exact time the turbidity start in the solution.
Can the author perform UV-Vis scattering plot with time to get information about these different times. Furthermore, can such measurements also give indication about the gelation time. In fact, scattering plot should also be able to show a co-operative self-assembly process.
10. If I understand it well, t_{vis} should be defined as the "time required for the gel to grow from the nucleus into the smallest macroscopic size discernible by the naked eye.
11. Fig. 8 caption says log-log plot but it seems to be just a log plot. The numbers written in the b and c i.e. 8.36 and 3.86 should be also defined-described in the figure caption.

12. Supplementary fig. 2: Considering the large variation in the gelation time reported in Table 1, how reliable is this? How many samples were measured?

13. Supplementary Fig. 15: The graph on the left panel is repeated three times in a,b and c but not sure what conc. data is this.

14. Examples of some typographical error " even without labeling to molecules", " not hierarchically", "revealed dintinguishable", "This would more spatiotemporal control of gel formation"

Reviewer #2

(Remarks to the Author)

The manuscript by Kimura et al. reports the investigation of supramolecular gelation by high-speed atomic force microscopy (HS-AFM). Previously, HS-AFM has been used to visualize dynamics of proteins and self-assembly of synthetic molecules, and the present study has further extended its use for elucidating more complex dynamic behavior: that is, supramolecular gel formation. Interestingly, the authors captured an intermittent growth with repeated elongation and pause phases during the formation of mesoscopic fibers, which are thicker than so-called supramolecular polymers. The authors proposed a block-stacking model to explain this unique propagation dynamics. In the latter part of the manuscript, the authors discussed gelation process based on macroscopic observation by naked eyes in relation to microscopic nucleation, thereby estimating the size of the nucleus and proposing the possible gelation mechanisms.

Although supramolecular gels have been studied to date extensively, as the authors point out in the introduction, its mechanism has remained still elusive. The present study shows interesting results and provides insightful discussion, which I think inspire many researchers in this field to tackle the mechanistic studies of supramolecular gelation. Consequently, designing supramolecular gels in terms not only of its monomer structures but also macroscopic time-evolution and physical properties would become possible. Overall, the manuscript is clearly written and insightful, as such I recommend its publication in Nature Communications, after the authors address the following points.

1) HS-AFM revealed that the growth rates of the two fiber ends were considerably different, which was explained based on the polarity of the fibers originating from the arrays of hydrogen-bonded urea units (Figure 6a). However, if the 1D arrays of hydrogen-bonded urea units were arranged antiparallel to each other, as indicated by the X-ray analysis for the UCCy, fibers should not have such polarity, and both the two fiber ends are equivalent. Did the authors observe such difference in the growth rates for other fibers? In addition, the authors estimate the distributions of rotational isomers (A, B, and C) are one-thirds, when explaining the difference of the growth rates of the two fiber ends. Can the author estimate the distributions using, for example, NMR and calculation (Figure S4)? If the authors assumption that the rotational isomerization is the rate determining step is correct, the rotational isomerization should be slower than the fiber growth rate, which I expect to be compatible with or slower than the NMR time scale.

2) In page 14, it is written that These analyses indicate that aggregate formation proceeds with the octamer and tetramer as the nuclei. For me, it was not very clear whether the authors consider that there are two possible nuclei size (octamer and tetramer) or that the nuclei size has size distribution between the octamer and tetramer.

3) It would be helpful if the authors define t_{nuc} and t_{vis} in Figure 10.

Reviewer #3

(Remarks to the Author)

Reviewer #4

(Remarks to the Author)

A delightful manuscript to read. The authors have meticulously described and investigated the gelation phenomena of a urea-based gelator relating to describing its self-assembly driven through nucleation processes. The use of HS-AFM has provided some fantastic insights into the numerous steps such gelators can go through. I can only but recommend the publication of the manuscript. I give some small recommendations to try and help improve some aspects of the paper.

The use of crystal structures and their analysis to infer structure motifs/synthons of the supramolecular fibres is a well-taken path within the supramolecular gel community but/and is fraught with mistakes and misunderstandings. The authors have covered this very well and used the crystal data appropriately. Two related things I felt the authors did not quite cover are:

1. the literature (evidence) of the urea tape hydrogen bonding isn't significant enough to "nail" home the proof of urea assembly and I would recommend adding more discussion and citation of work covering urea assembly (a good place to start would be to cover works by Custelcean, Diaz, Steed, as examples).

2. Observing such large and regular structures from a small molecule's assembly has to make the reader go, how is this different to crystallisation and are they crystalline? The history of crystal growth and the literature of it is significantly vast and when reading such papers it is sometimes noted that the field of supramolecular polymers is reinventing the wheel, somewhat harshly stating so but is factually correct. Invoking periodic assembly in their models' descriptions obviously

means the comparison in terms of the physical laws is likely to be needed and justified (how many times have we seen the mathematics repeated when describing similar phenomena from different aspects of science, nucleation and autocatalysis as examples). So with those discussions, I would like to know if the authors have some other evidence of the structural nature of their structures? Have they done any scattering (PXRD, SAXS, SANS, WAXS etc) to observe any periodic structure and/or secondary evidence of the sizes of their assemblies in situ (Adams and Terech/Weiss have some excellent literature on this)? Are CD, IR, Raman also appropriate tools to find potential correlations between the conformations and the observed structures?

I feel more expansion of the statement is needed for Line 364-365, “, as the existing fiber can catalyse it”. I have had a particular interest in the concept of supramolecular polymers self-catalysing (induction) new polymers/fibres/fibrils. As this is closely linked to the concepts of nucleation and growth dynamics. I have yet to find a good description (based on theoretical or analytical methods) of why we all observe this “catalysis” of fibres. Especially given observations varying from “doing nothing (only convection and/or Brownian motion acting mechanically on structures?)” to sonication to “autocatalysis”. So I would recommend the authors add some content to this statement with some relevant literature to discuss it in the context of their models and observations.

Version 1:

Reviewer comments:

Reviewer #1

(Remarks to the Author)

The authors have satisfactorily addressed the concerns raised by me. Therefore, I recommend its publication in the current form.

Furthermore, as requested by the editor, I have also looked over the responses of the authors to the concerns from Reviewer #4 as well as evaluated their responses. They seem to be satisfactory enough to be accepted for publication.

Mohit

Reviewer #2

(Remarks to the Author)

The authors have addressed properly the comments from the reviewers. I think that the manuscript has been improved to read clearly. This study provides new insights into supramolecular gelation, and I recommend it to be published in Nature Communications in the revised form.

Reviewer #3

(Remarks to the Author)

Response to the Comments of Reviewer #1

(Revisions to the main text and Supplementary Information are highlighted yellow)

General Comment

The article by Kimura et al. presents molecular insights into the widely studied supramolecular gelation process using high-speed AFM imaging performed in the native liquid phase. This work chooses a known urea derivative to study the self-assembly and gelation and provide unique understanding into the transformation from nanoscopic to microscopic and macroscopic scale. They show asymmetrical growth kinetics of the two end of the fibers, and repeated elongation-pause phase in the fiber growth, three steps involved in the gelation and the substrate induced effects in the self-assembly. The authors finally propose a theoretical block-stacking model to understand the observation. Overall, the manuscript is one of the very few examples of supramolecular fiber formation probed with nanoscale details in solution and provides some useful information on the process of self-assembly of the chosen molecule. **However, the conclusions drawn from the work is specifically relevant to this molecule and it cannot be generalized. Furthermore, the substrate effect seems to play some role and thus it may not be obvious to extrapolate all observations to the processes happening in the bulk.** Also, some conclusions drawn are not fully convincing. The reviewer feels that the manuscript is not easy to read and understand because of the lack of some information (described below) at different places. Thus, I do not support its publication in Nature Communication, at least not in its current form.

Response to General Comment

Thank you for your important comment. We would like to respond to the important underlined parts of the above General Comments. The generality of the findings obtained from the research is an important aspect. The self-assembly of **UC13**, the gelator molecules in this study is based on "hydrogen bonds", aromatic interactions, and van der Waals forces between alkyl chains, which are the driving forces for the formation of general gel networks or supramolecular polymers, and we believe that the findings of this study have a significant impact on many of these supramolecular materials.

In addition, it is certainly necessary to consider the effect of the substrate when observing using a microscope. This has a particularly significant effect on the aggregates (at the lowest level) that grow while directly interacting with the substrate. In this study, HS-AFM measurements showed that the fibrils in the first and second steps grew along the HOPG lattice, indicating that they were affected by the substrate (Figure 3). On the other hand, the thick fibers observed in the third step, which are the important stage for gelation, were shown to be not affected by the substrate. Since these fibers are sufficiently stabilized in the lateral direction by van der Waals forces and aromatic interactions, it is thought that they can grow without influence from the substrate. Therefore, we concluded that the thick fibers required for gelation are not formed by the bundling of thin fibrils in a hierarchical manner, but rather the existing fibers formed through stabilization by multiple interactions grow.

Specific Comment 1

Would the author be able to visualize any other class of gels, even the very simplest known ones, or their method is specific to this gel?

Response to Specific Comment 1

One of the authors (T. Uchihashi) has already studied many other gel-forming supramolecular fibers, and one of the works has been published in revised ref. 40. Also T. Uchihashi and S. Yagai have already succeeded to visualize the growth of gel-forming supramolecular fibers of barbituric-acid hydrogen-bonding monomers (newly added ref. 43). Accordingly, we are sure that our method based on HS-AFM observation could be a general method to scrutinize the mechanism of supramolecular gel formation. Also, recently HS-AFM observation of supramolecular fibers was also reported by other groups (revised ref. 41, 42). We would like to emphasize the importance of the HS-AFM technique to understand molecular-level mechanism of supramolecular self-assembly phenomena, which is important to develop smart materials science. Based on your comments, we added an additional reference to the main text.

Newly added references

43. Tamaki, K., Datta, S., Hanayama, H., Ganser, C., Uchihashi T. & Yagai S. Photoresponsive supramolecular polymers capable of intrachain folding and interchain aggregation. *J. Am. Chem. Soc.* **146**, 22166–22171 (2024). (page 5, 21, highlighted with yellow)

Specific Comment 2

In figure 1b, octameric nucleus seems to be made of 8 monomers, but in the manuscript authors repeatedly mention "It should be noted that this octamer nucleus is not necessarily an isolated octamer in the solution; it might be an octameric substructure within a larger oligomer."

Response to Specific Comment 2

Thank you for your comment. The high probability is that the octameric nucleus exists as an isolated octamer in solution, as shown in Figure 1b. However, it is also possible that an octameric substructure is embedded in a larger aggregate. Considering your comment, we revised the text as follows.

Original sentence: This exponent indicates a nuclear size of 8 (or 6–10, considering statistical uncertainty), although the exponent must be interpreted carefully in terms of the concentration dependence using the nuclear size.⁶⁸⁻⁷⁰ It should be noted that this octamer nucleus is not necessarily an isolated octamer in the solution; it might be an octameric substructure within a larger oligomer.

Revised sentence: This exponent indicates a nuclear size of 8 (or 6–10, considering statistical uncertainty). While this octamer nucleus may exist as an isolated octamer in solution, as depicted in Figure 1b, another possibility has been pointed out in the literature⁶⁵⁻⁶⁸ regarding the interpretation of concentration dependence by nucleus size. The alternative suggests an octameric substructure embedded in a larger aggregate, with the nucleation corresponding to the formation of such substructures through conformational conversion inside the aggregate. This possibility becomes particularly relevant when interpreting non-integer exponents.⁶⁷ (page 14, highlighted with yellow)

Specific Comment 3

"At the end of the HS-AFM observation (after 40 min), macroscopic supramolecular gel formation was observed..." This is confusing considering that in Table 1 we see the gelation time in weeks and here it is happening in 40 min.

Response to Specific Comment 3

Thank you for your comment. Since the experimental procedures for the gelation experiment and HS-AFM experiment were different, they are not directly comparable. Firstly, the sample volume was different (500 μL vs 70 μL , respectively). Secondly, the gelation experiments were carried out in vials, while the HS-AFM experiments were carried out on a substrate. In the gelation experiment, the mixture of **UC13** and DMSO was heated to dissolve, and then cooled to room temperature to form a gel (Supporting Information). In the HS-AFM experiments, gelator solutions were added to DMSO solvent, and the adsorption of molecules and growth behavior of supramolecular fibers on the HOPG substrate were observed. Particularly, it was observed that the solvent added beforehand significantly shortened the gelation time in the HS-AFM experiment using DMSO.

Original sentence: At the end of the HS-AFM observation (after 40 min), macroscopic supramolecular gel formation was observed at the fluid reservoir within the cantilever holder (Figure 2e).

Revised sentence: At the end of the HS-AFM observation (after 40 min), gelation of the specimen solution was observed (Figure 2e). (page 7, highlighted with yellow)

Newly added sentence in Supplementary Methods (High-speed atomic force microscopy)

(The total volume of the solvent used was 70 μL). (page S3, highlighted with yellow)

Specific Comment 4

"The low gelation ability of UC13 in DMSO (minimum gelation concentration: 30–50 mM) is not suitable for the HS-AFM observation of much mesoscopic fibers inducing physical gelation." Not clear why is it so.

Response to Specific Comment 4

Thank you for your comment. Our original sentence was not clear. We wanted to say that gelation in DMSO occurs more instantly, making it difficult to capture the dynamic formation process of supramolecular fibers at the early stage of gelation. Therefore, we revised the sentence as follows.

Original sentence: The low gelation ability of **UC13** in DMSO (minimum gelation concentration: 30–50 mM) is not suitable for the HS-AFM observation of much

mesoscopic fibers inducing physical gelation. As an alternative low volatility solvent suitable for HS-AFM observation, we used an ionic liquid (EMI-Tf₂N) that could be gelled at much lower concentrations (1.8–5 mM).

Revised sentence: Because the minimum gelation concentration of **UC13** in DMSO is very high (30–50 mM), gelation occurs more instantly, making it difficult to capture the dynamic formation process of supramolecular fibers at the early stage of gelation.

As an alternative low volatility solvent suitable for HS-AFM observation, we used an ionic liquid (EMI-Tf₂N) that could be gelled at much lower concentrations (1.8–5 mM). (page 7, highlighted with yellow)

Specific Comment 5

" In the first step, the two-dimensional (2D) growth of unimolecular-thickness sheets was observed" How is this deciphered? Authors should consider providing the height color scale of this AFM image and the topographical height plot.

Response to Specific Comment 5

We apologize for confusing the reviewers. As stated in the caption, the HS-AFM images in Figures 3 and 4 are phase images, which do not contain height information. Nevertheless, the sheet-like structure observed in the first step was described as "unimolecular-thickness sheets". Therefore, we revised the sentence as follows.

Original sentence: In the first step, the two-dimensional (2D) growth of unimolecular-thickness sheets was observed on HOPG substrate several minutes after adding the gelator into the solution (Figure 3a).

Revised sentence: In the first step, the two-dimensional (2D) growth of **molecular sheets** was observed on HOPG substrate several minutes after adding the gelator into the solution (Figure 3a). (page 7, highlighted with yellow)

Due to the reasons mentioned above, height information cannot be displayed as color scale. However, in response to your comment, we showed the corresponding topographic (height) images in **Rev-only-Fig. 1**. The thickness of the sheet is less than 1 nm, causing poor contrast.

Rev-only-Fig. 1.

We also analyzed the height information in Figure 2c, which is shown in **Rev-only-Fig. 2.**

Rev-only-Fig. 2.

Since the reason for using the phase image had only been described in the Supplementary Methods, we added it to the Methods.

Newly added sentence in Methods (High-speed atomic force microscopy)

Phase images often provide better contrast and sharper images than topographic images in fiber imaging (Supplementary Fig. 1). Therefore, the discussion in this

article has been based on phase images, unless stated otherwise. (page 17, highlighted with yellow)

Specific Comment 6

"second step involved the growth of short fibrils with a triangular pattern" Author should consider highlighting the triangular pattern on the image in Figure 3b.

Response to Specific Comment 6

Thank you for your helpful comment. We highlighted the triangular pattern in Figures 3a and b.

Original Figure 3:

Figure 3 | HS-AFM observation of UC13 in EMI-Tf₂N. a–c, Clipped phase-contrast HS-AFM images of the fibrillation process of the 2.0 mM **UC13** in EMI-Tf₂N on the HOPG surface: **a**, First step: the 2D growth of unimolecular-thickness sheet; **b**, Second step: the growth of short fibrils and the complete coverage of the substrate; **c**, Third step: the elongation of thick bundled fibers. Scale bar: 100 nm. Imaging rate: (a & b) 1 and (c) 2 s/frame. See also Supplementary Movie 2.

Revised Figure 3:

Figure 3 | HS-AFM observation of UC13 in EMI-Tf₂N. **a–c**, Clipped phase-contrast HS-AFM images of the fibrillation process of the 2.0 mM **UC13** in EMI-Tf₂N on the HOPG surface: **a**, First step: the 2D growth of unimolecular-thickness sheet; **b**, Second step: the growth of short fibrils and the complete coverage of the substrate; **c**, Third step: the elongation of thick bundled fibers. Scale bar: 100 nm. Imaging rate: (a & b) 1 and (c) 2 s/frame. See also Supplementary Movie 2. Representative time-lapse snapshots showing similar results from at least three independent experiments are presented. (page 27, The gray highlights mean revisions in accordance with the formatting instructions.)

Specific Comment 7

Can such imaging be performed on other substrate like Mica?

Response to Specific Comment 7

UC13 did not adsorb to the mica surface and could not be observed. Although we also tried using amino silane-treated mica, **UC13** did not adsorb. We consider that the hydrophobicity of the substrate surface is important for adsorption of **UC13**.

Specific Comment 8

Authors suggest the possibility of " π - π or possibly CH- π ". Does it show any change in the UV-Vis absorption spectra?

Response to Specific Comment 8

Thank you for addressing this issue. In the crystal structure of **UCy**, the 2-benzylphenyl groups are positioned far from each other for π - π stacking. In addition, there was no significant change in the UV-Vis absorption spectra. On the other hand, the optimized structure of the parallel dimer of **UC13**, obtained using DFT calculations, suggests the potential presence of CH/ π interactions. Thus, we revised main text as follows.

Original sentence: The direct growth of 20 nm-wide fibers suggests that fiber growth was driven not only by urea-urea hydrogen-bonding along the fiber axis, but also by the fiber-stabilizing van der Waals interactions between the alkyl chains and π - π or possibly CH- π interactions between the benzyl units in the lateral (horizontal) direction.

Revised sentence: The direct growth of 20 nm-wide fibers suggests that fiber growth was driven not only by urea-urea hydrogen-bonding along the fiber axis, but also by the fiber-stabilizing van der Waals interactions between the alkyl chains or possibly CH- π interactions in the lateral (horizontal) direction (newly added Supplementary Fig. 13). (page 10, highlighted with yellow)

Supplementary Figure 13. Optimized structure of the parallel dimer of **UC13**. (page S16, highlighted with yellow)

Newly added Supplementary Methods (Conformational search for **UC13**)

Interatomic distance between UC13s in dimer: For the purpose of clarifying the intermolecular structure between **UC13s**, geometry optimization of **UC13** dimer was

performed at the B3LYP-D3/6-311+G(2d,p) level (newly added ref. 15) using Gaussian 16 program (ref.14). The initial geometry of the dimer was constructed based on the monomer of conformation No. 4; the one is set to the original coordinates of monomer, and the other is that y and z coordinates are shifted from the original by -4\AA (newly added Supplementary Fig. 3). (page S5, highlighted with yellow)

Supplementary Fig. 3: Initial geometry of **UC13** dimer for optimization. (page S5, highlighted with yellow)

Newly added references in Supplementary Methods

15. Grimme, S. Antony J. Ehrlich S. & Krieg H. A consistent and accurate ab initio parameterization of density functional dispersion correction (DFT-D) for the 94 elements H-Pu. *J. Chem. Phys.* 132, 154104 (2010). (page S23, highlighted with yellow)

Specific Comment 9

" This implies that the supramolecular gel formation is dominated by the formation of a single nucleus in the entire solution." This is a very bold statement, and the supporting data is not very convincing. Firstly, this reviewer believes that manual and visual probing of the tstart and tvis is prone to a lot of errors as is also evident. Its difficult to manually point out at what exact time the turbidity start in the solution. Can the author perform UV-Vis scattering plot with time to get information about these different times. Furthermore, can such measurements also give indication

about the gelation time. In fact, scattering plot should also be able to show a co-operative self-assembly process.

Response to Specific Comment 9

We appreciate your critical comments. As you pointed out, if we could quantitatively evaluate the gelation times with UV-Vis scattering plot rather than observing with the naked eye, we would be able to provide more precise discussion. However, the preparation of supramolecular gels requires heating the **UC13**-solvent mixture over 100 °C, which makes these experiments difficult to be performed in conventional cuvettes. An alternative method is to prepare a hot solution of **UC13** in a vial, and transfer it to the cuvette. However, we found that such a procedure could affect lag-phase prior to gelation significantly. Based on these reasons, we decided to employ visual inspection to judge the onset of gelation.

On the other hand, we can understand the reviewer's concern. We thus performed image analysis on the movie files displaying time-evolution of turbidity changes. The image analysis revealed that there was no significant difference between the results obtained by the naked eye and those obtained from the image analysis. The main text, Figure 8, and Supplementary Table 3 were revised according to the results of the image analysis, and the method of image analysis was described in Supplementary Methods.

Original sentence: In a solution without aggregates, molecules exist in a supersaturated state. When the formation of nuclei, which serves as templates for the subsequent growing process, occur, aggregate formation propagates throughout the system. To shed light on the nucleation process, we studied the concentration dependence of the lag time in the supramolecular gel formation of **UC13** in EMI-Tf₂N. A closer inspection of the gelation process in a vial by the naked eye revealed that the turbidity caused by the aggregation of **UC13** occurred at a single point in the homogeneous solution, and gelation was completed when the turbidity had spread throughout the system (Figure 8a and Supplementary Movie 8). Therefore, we defined the gelation start time (t_{start}) as the time when the turbidity occurred visibly, and the gelation completion time (t_{comp}) as the time when the turbidity had spread throughout the system. t_{start} and t_{comp} were determined as the average of ten or more experiments (Supplementary Table 3). For the 2.2 mM solution, t_{start} and t_{comp} were 224 ± 38 min and 626 ± 121 min, respectively, which became shorter ($t_{\text{start}} = 27 \pm 7.9$ min; $t_{\text{comp}} = 87.5 \pm 19$ min) upon a concentration increase to 3.0

mM. In contrast, t_{start} and t_{comp} became longer to 657 ± 190 min and $1,168 \pm 178$ min, respectively, upon a concentration decrease to 1.8 mM. At concentrations below 1.5 mM, no supramolecular gel formation was observed, and a suspension of fibrous macroscopic aggregates was obtained instead (Supplementary Fig. 13). At 1.0 mM, no macroscopic aggregate formation was observed, and the solution remained homogeneous.

Revised sentence: In a solution without aggregates, molecules exist in a supersaturated state. When the formation of nuclei, which serves as templates for the subsequent growing process, occur, aggregate formation propagates throughout the system. To shed light on the nucleation process, we studied the concentration dependence of the lag time in the supramolecular gel formation of **UC13** in EMI-Tf₂N. A closer inspection of the gelation process in a vial by **movie** revealed that the turbidity caused by the aggregation of **UC13** occurred at a single point in the homogeneous solution, and gelation was completed when the turbidity had spread throughout the system (Figure 8a and Supplementary Movie 8). Therefore, we defined the gelation start time (t_{start}) as the time when the turbidity occurred visibly, and the gelation completion time (t_{comp}) as the time when the turbidity had spread throughout the system. **Details of the algorithm used to determine t_{start} and t_{comp} from digitally recorded movies are given in Supplementary Information. Statistics of ten samples were taken for each concentration and the average and standard deviation of t_{start} and t_{comp} were obtained (Supplementary Table 3).** For the 2.2 mM solution, t_{start} and t_{comp} were 311 ± 67 min and 604 ± 117 min, respectively, which became shorter ($t_{\text{start}} = 97 \pm 8$ min; $t_{\text{comp}} = 155 \pm 8$ min) upon a concentration increase to 2.8 mM. In contrast, t_{start} and t_{comp} became longer to 750 ± 205 min and $1,077 \pm 248$ min, respectively, upon a concentration decrease to 1.8 mM. At concentrations below 1.5 mM, no supramolecular gel formation was observed, and a suspension of fibrous macroscopic aggregates was obtained instead (Supplementary Fig. 16). At 1.0 mM, no macroscopic aggregate formation was observed, and the solution remained homogeneous. (page 12, highlighted with yellow)

Original sentence: According to this figure, $\langle t_{\text{nuc}} \rangle$ has a power-law dependence on the monomer concentration, with an estimated exponent of 8.36 ± 0.86 , where the uncertainty is the 1σ error estimated using the bootstrap method.⁶⁷

Revised sentence: According to this figure, $\langle t_{\text{nuc}} \rangle$ has a power-law dependence on the monomer concentration, with an estimated exponent of 7.77 ± 0.69 , where the uncertainty is the 1σ error estimated using the bootstrap method (revised ref. 64). (page 13, highlighted with yellow)

Original sentence: However, given the statistical uncertainty, the present estimate of 8.36 ± 0.86 for the exponent is not sufficiently statistically accurate to conclude a non-integer power. Having obtained $\langle t_{\text{nuc}} \rangle$, we also evaluated t_{vis} by subtracting $\langle t_{\text{nuc}} \rangle$ from $\langle t_{\text{start}} \rangle$ in equation (1). Figure 8c shows the plot of $\langle t_{\text{vis}} \rangle$ versus the monomer concentration. The time follows a power-law dependence with an exponent of 3.86 ± 0.29 .

Revised sentence: However, given the statistical uncertainty, the present estimate of 7.77 ± 0.69 for the exponent is not sufficiently statistically accurate to conclude a non-integer power. Having obtained $\langle t_{\text{nuc}} \rangle$, we also evaluated t_{vis} by subtracting $\langle t_{\text{nuc}} \rangle$ from $\langle t_{\text{start}} \rangle$ in equation (1). Figure 8c shows the plot of $\langle t_{\text{vis}} \rangle$ versus the monomer concentration. The time follows a power-law dependence with an exponent of 3.95 ± 0.27 . (page 14, highlighted with yellow)

Original Figure 8:

Figure 8 | Relationship between gelation time and concentration of UC13. **a**, Definition of t_{start} and t_{comp} . **b**, Log-log plot of the nucleation time vs. concentration obtained from macroscopic observation of gel formation. **c**, Log-log plot of the time required for the gel to grow into a visible size vs. concentration obtained from the observation of macroscopic gel formation. See also Supplementary Movie 8.

Revised Figure 8:

Figure 8 | Relationship between gelation time and concentration of UC13. a, Definition of t_{start} and t_{comp} . See also Supplementary Movie 8. b, Log-log plot of the nucleation time vs. concentration obtained from macroscopic observation of gel formation. Fitted line to the power law $\langle t_{\text{nuc}} \rangle \propto [\text{UC13}]^{-\alpha}$, which becomes a straight line $\log \langle t_{\text{nuc}} \rangle = -\alpha \log [\text{UC13}] + \text{const.}$ in the log-log plot, is shown with the value of the exponent α . Error bars show 95 % confidence intervals estimated by the bootstrap method. The uncertainties in the fitted exponents are the 1σ error estimated by the bootstrap method. Source data of the graphs are provided as a Source Data file. c, Log-log plot of the time required for the gel to grow into a visible size vs. concentration obtained from the observation of macroscopic gel formation. Fitted line to the power law $\langle t_{\text{nuc}} \rangle \propto [\text{UC13}]^{-\alpha}$, which becomes a straight line $\log \langle t_{\text{nuc}} \rangle = -\alpha \log [\text{UC13}] + \text{const.}$ in the log-log plot, is shown with the value of the

exponent α . Error bars show 95 % confidence intervals estimated by the bootstrap method. The uncertainties in the fitted exponents are the 1σ error estimated by the bootstrap method. Source data of the graphs are provided as a Source Data file. (page 34–35, highlighted with yellow)

Original Supplementary Table 3:

Supplementary Table 3. Relationship between the **UC13** concentration and the gelation time.*

Concentration (mM)	t_{start} (min)	t_{comp} (min)
3.2	20 ± 10	53 ± 23
3.0	27 ± 7.9	87.5 ± 19
2.8	91 ± 5.4	162 ± 6.0
2.4	119 ± 24	432 ± 72
2.2	224 ± 38	626 ± 121
2.0	377 ± 126	838 ± 206
1.8	657 ± 190	1,168 ± 178

*All runs were performed under same conditions.

Revised Supplementary Table 3:

Supplementary Table 3. Relationship between the **UC13** concentration and the gelation time.*

Concentration (mM)	t_{start} (min)	t_{comp} (min)
3.2	15 ± 0.5	24 ± 0.6
3.0	26 ± 4.5	57 ± 7.7
2.8	97 ± 7.5	155 ± 8.3
2.4	194 ± 38	390 ± 38
2.2	311 ± 67	604 ± 117
2.0	461 ± 173	735 ± 282
1.8	750 ± 205	1,077 ± 248

*All runs were performed under same conditions.

*For each concentration, the average and standard deviation calculated from ten samples are shown. (page S22, highlighted with yellow)

Newly added Supplementary Methods

Time-course Analysis of Gelation: A mixture of **UC13** and EMI-Tf₂N (500 mL) in a glass vial bottle was heated on a hot plate (200 °C) until dissolved. Obtained solution

was gradually cooled to ambient temperature. Ten vials of the same solution were prepared for each concentration. Change of the turbidity in a vial was monitored by recording into digital movie by iPad Air (5th generation). Frames are extracted every one minute from the movie file, and image analysis to evaluate the turbidity was performed. In the first frame, two rectangular regions for each vial were assigned, one enclosing the vial and the other designating the solution region. In the subsequent frames, the vial rectangles are re-assigned through template-matching with the image in the previous frame by evaluating the zero-mean normalized cross-correlation. The position of the solution rectangle is accordingly re-assigned by keeping the relative position from the vial rectangle. Turbidity was evaluated by calculating the average brightness of the pixels contained in the solution rectangle. To minimize the effect of the room illumination, the average brightness in the upper half of the vial rectangle was subtracted as background. Then, the obtained turbidity values as a function of time were fitted to the following analytic formula:

$$f(t) = b + \frac{a}{1 + \exp(-k(t - t_c))}$$

where t is time, and the parameters a , b , t_c , and k were estimated by least-squares fitting. In this formula, the turbidity values before and after the gelation are given by $f(-\infty) = b$ and $f(+\infty) = a + b$. Then the gelation start time t_{start} was defined as $t_{\text{start}} = t_c - 2a/k$. This is the time at which the line tangent to $y = f(t)$ at $t = t_c$ cuts the horizontal line $y = b$. Similarly, the gelation completion time t_{comp} was defined as $t_{\text{comp}} = t_c + 2a/k$, the time at which the same tangent line cuts $y = a + b$. (page S2, highlighted with yellow)

Specific Comment 10

If I understand it well, t_{vis} should be defined as the "time required for the gel to grow from the nucleus into the smallest macroscopic size discernible by the naked eye.

Response to Specific Comment 10

Thank you for your useful comment. It was revised.

Original sentence: where t_{nuc} is the time for nucleation and t_{vis} is the time required for the gel to grow into the smallest macroscopic size discernible by the naked eye.

Revised sentence: where t_{nuc} is the time for nucleation and t_{vis} is the time required for the gel to grow from the nucleus into the smallest macroscopic size discernible by the naked eye. (page 13, highlighted with yellow)

Specific Comment 11

Figure 8 caption says log-log plot but it seems to be just a log plot. The numbers written in the b and c i.e. 8.36 and 3.86 should be also defined-described in the figure caption.

Response to Specific Comment 11

Thank you for your comment. It is correct that Figures 8b and c are log-log plots. Based on your comment, we added explanations to the Figure 8 captions for the values 8.36 and 3.86 (Please refer to **Response to Specific Comment 9** for the revised sentences).

Specific Comment 12

Supplementary fig. 2: Considering the large variation in the gelation time reported in Table 1, how reliable is this? How many samples were measured?

Response to Specific Comment 12

Thank you for your comment. As Supplementary Fig. 2 and Supplementary Table 2 are the conformational analysis of **UC13**, we think you mean Supplementary Table 1 or Supplementary Table 3. The number of samples is 20 for Supplementary Table 1, and it is 10 for Supplementary Table 3. As the number of samples was lacking in Supplementary Table 3, we added it (Please refer to **Response to Specific Comment 9** for the revised sentences).

Specific Comment 13

Supplementary Fig. 15: The graph on the left panel is repeated three times in a,b and c but not sure what conc. data is this.

Response to Specific Comment 13

Thank you for pointing this out. Supplementary Figure 15 was revised based on your comments (Revised Supplementary Fig. 18). There were no changes to the main text, but the values in the histograms in Revised Supplementary Fig. 18 were revised as a result of the reanalysis.

Original Supplementary Figure 15:

Supplementary Figure 15. Growth rate and duration of the fiber in the third step. **a**, Fiber elongation over time (left); Elongation rates in the area marked by the dotted lines on the distance vs time plots on the left (center and right). The black curve shows Gaussian fitting, and the elongation rate was calculated from the median value. The elongation rates were 17, 16, 18 nm/s for the **UC13** concentrations of 1.5, 2.0, and 3.0 mM, respectively. **b**, Fiber elongation over time (left); Elongation durations highlighted by the green areas on the distance vs time plots on the left (center and right). The black curves show a single exponential function fitting. The elongation durations were 1.1, 0.9, and 1.0 s at 1.5, 2.0, and 3.0 mM, respectively. **c**, Fiber elongation over time (left); Pause durations highlighted by the blue areas on the distance vs time plots on the left (center and right). The black curves show a single exponential function fitting. Pause durations were 5.9, 6.0, and 6.0 s at 1.5, 2.0, and 3.0 mM, respectively.

Revised Supplementary Figure 15:

Supplementary Figure 18. Growth rate and duration of the fiber in the third step. The graph on the left in Supplementary Fig. 18a–c shows the growth rate of typical fibers (3.0 mM). In Supplementary Fig. 18a, the growth rate of the area circled by the dotted line was analyzed. In Supplementary Fig. 18b, the times of the area highlighted by the green square were measured. In Supplementary Fig. 18c, the times of the area highlighted by the blue square were measured. **a**, Typical fiber elongation over time (left, 3.0 mM); Elongation rates in the area marked by the dotted lines on the distance vs time plots on the left (center and right). The black curve shows Gaussian fitting, and the elongation rate was calculated from the median value. The elongation rates were 17, 16, 18 nm/s for the **UC13** concentrations of 1.5 ($n = 132$), 2.0 ($n = 116$), and

3.0 ($n = 99$) mM, respectively. **b**, Typical fiber elongation over time (left, 3.0 mM); Elongation durations highlighted by the green areas on the distance vs time plots on the left (center and right). The black curves show a single exponential function fitting. The elongation durations were 1.1, 0.9, and 1.0 s at 1.5 ($n = 132$), 2.0 ($n = 104$), and 3.0 mM ($n = 96$), respectively. **c**, Typical fiber elongation over time (left, 3.0 mM); Pause durations highlighted by the blue areas on the distance vs time plots on the left (center and right). The black curves show a single exponential function fitting. Pause durations were 5.9, 6.0, and 6.0 s at 1.5 ($n = 133$), 2.0 ($n = 86$), and 3.0 mM ($n = 85$), respectively. Representative analysis was performed on one fiber out of many fibers that showed similar results. Source data of the histograms are provided as a Source Data file. (page S20–21, highlighted with yellow)

Specific Comment 14

Examples of some typographical error " even without labeling to molecules", " not hierarchically", "revealed dintinguishable", "This would more spatiotemporal control of gel formation"

Response to Specific Comment 14

Thank you for pointing out. The errors were revised. Please refer to page 5, 9, 10, 16.

Response to the Comments of Reviewer #2

(Revisions to the main text and Supplementary Information are highlighted green)

General Comment

The manuscript by Kimura et al. reports the investigation of supramolecular gelation by high-speed atomic force microscopy (HS-AFM). Previously, HS-AFM has been used to visualize dynamics of proteins and self-assembly of synthetic molecules, and the present study has further extended its use for elucidating more complex dynamic behavior: that is, supramolecular gel formation. Interestingly, the authors captured an intermittent growth with repeated elongation and pause phases during the formation of mesoscopic fibers, which are thicker than so-called supramolecular polymers. The authors proposed a block-stacking model to explain this unique propagation dynamics. In the latter part of the manuscript, the authors discussed gelation process based on macroscopic observation by naked eyes in relation to microscopic nucleation, thereby estimating the size of the nucleus and proposing the possible gelation mechanisms.

Although supramolecular gels have been studied to date extensively, as the authors point out in the introduction, its mechanism has remained still elusive. The present study shows interesting results and provides insightful discussion, which I think inspire many researchers in this field to tackle the mechanistic studies of supramolecular gelation. Consequently, designing supramolecular gels in terms not only of its monomer structures but also time-evolution and physical properties would become possible. Overall, macroscopic the manuscript is clearly written and insightful, as such I recommend its publication in Nature Communications, after the authors address the following points.

Response to General Comment

We appreciate your thoughtful and encouraging comments. We hope that our research will lead to a significant breakthrough in elucidating the mechanisms of supramolecular gels and enable the rational control of their physical properties and functions.

Specific Comment 1

HS-AFM revealed that the growth rates of the two fiber ends were considerably different, which was explained based on the polarity of the fibers originating from the arrays of hydrogen-bonded urea units (Figure 6a). However, if the 1D arrays of hydrogen-bonded urea units were arranged antiparallel to each other, as indicated by the X-ray analysis for the UCCy, fibers should not have such polarity, and both the two fiber ends are equivalent. Did the authors observe such difference in the growth rates for other fibers? In addition, the authors estimate the distributions of rotational isomers (A, B, and C) are one-thirds, when explaining the difference of the growth rates of the two fiber ends. Can the author estimate the distributions using, for example, NMR and calculation (Figure S4)? If the authors assumption that the rotational isomerization is the rate determining step is correct, the rotational isomerization should be slower than the fiber growth rate, which I expect to be compatible with or slower than the NMR time scale.

Response to Specific Comment 1

We appreciate your important comment. The phenomenon that the growth rate differs at the two ends of the fiber was also observed in other fibers as well as the one we focused on in Figure 5a, and thus this phenomenon is general for **UC13**. We have not yet succeeded in obtaining crystals of **UC13**, which has a linear alkyl group. The crystal structure shown in Figure 2d and revised Supplementary Fig. 6 is that of **UCCy**, which has a cyclohexyl group. The crystal structure analysis of **UCCy** was performed to confirm that urea forms a one-dimensional hydrogen-bonded supramolecular polymer. We think that highly directional interactions such as urea and amide hydrogen bonding are the basis for the formation of molecular aggregates such as supramolecular gels and crystals, and that although the compounds are different, urea forms one-dimensional hydrogen bonded supramolecular polymers. On the other hand, the packing of the supramolecular polymers is not necessarily consistent between supramolecular gels and crystals. We think that metastable supramolecular gels (and supramolecular polymers) are different from crystal structures. We revised the main text and Figure 6 as follows to clarify our assertion that the anisotropic growth of the fibers is attributed to the self-assembly of **UC13** in a parallel manner. We also assume that the degree of the conformational freedom of the fiber end, rather than the monomer, may be responsible for anisotropic growth, and we added this to the main text. NMR experiments cannot be performed using EMI-Tf₂N, the solvent used for gelation.

Original sentence: The growth of the thick fibers observed in the third step was much slower than that in the preceding steps. The time evolution of fiber growth is shown in Figure 5b. The slope of the curve suggested an average growth rate of 0.07 nm/s. Interestingly, the kymograph of a single fiber (indicated by arrows in Figure 5a) revealed that the growth rates of the two fiber ends were considerably different, as the average growth rate of L-end of 0.06 nm/s while the R-end of 0.01 nm/s (Figure 5c and Supplementary Fig. 8). This finding reflects the unique anisotropic nature of urea supramolecular building blocks. According to the generally employed model for hydrogen-bonded ureido groups, the resulting chain ends should expose either C=O or N–H groups (Figure 6a). This model was confirmed by the aforementioned single-crystal X-ray diffraction analysis of **UCCy** (Figure 2d and Supplementary Fig. 5). Considering the binding of free monomers to both ends of the fiber, the concentrations of bindable conformational isomers depend on the rotation of the monomer around the two single bonds connecting the C=O and N–H groups (Figure 6b). The concentration of C=O groups binding to the N–H end does not depend on the conformational isomerism and equals the concentration of free monomers. Conversely, only the rotational isomer **B**, with two NHs pointing in the same direction, can bind to the C=O end; even if only the three planar conformations in Figure 6b are assumed, the concentration of this bindable conformational isomer is statistically only one third of that of the free monomers. Accordingly, the growth rate at the N–H end should be statistically three-fold faster than that at the C=O end, which is in line with the unequal growth rate at the both fiber ends.

Revised sentence: In the third step, the growth of the thick fibers was significantly slower compared to the preceding steps. The time evolution of fiber growth is shown in Figure 5b. The slope of the curve suggests an average growth rate of 0.07 nm/s. Interestingly, the kymograph of a single fiber (indicated by arrows in Figure 5a) revealed that the growth rates of the two fiber ends were markedly different. Specifically, the average growth rate at the L-end was 0.06 nm/s, whereas it was only 0.01 nm/s at the R-end (Figure 5c and revised Supplementary Fig. 10). This finding indicates that the hydrogen-bonded urea chains of **UC13** are organized in a parallel manner to form fibers (Figure 6a). This is because, in such supramolecular structures, the terminal functional groups of the fibers become C=O at one end and N–H at the opposite end. When considering the situation where free monomers bind to these fiber termini, the kinetic constant for this binding must depend on the rotation of the monomer around the two single bonds connecting the C=O and N–H groups (Figure

6b). The C=O groups binding to the N–H end do not depend on their orientation and are therefore equal to the concentration of free monomers. In contrast, only structure B, where the two NH groups point in the same direction, can bind to the C=O end. Even if only the three structures in Figure 6b are considered, the concentration of monomers capable of binding to the C=O end is statistically one-third of the free monomers. Consequently, the growth rate at the N–H end is statistically three times faster than that at the C=O end, which is consistent with the fact that the growth rates at the two fiber ends are unequal. On the other hand, if the rotational isomerization of the monomer occurs sufficiently rapidly, the degree of conformational freedom at the fiber termini may be responsible. Specifically, the C=O terminus remains structurally invariant, whereas the N–H terminus can undergo changes. In this case, contrary to the scenario described above, the growth rate at the C=O terminus could become faster. In either case, the rotational isomerization around the urea units is likely the origin of the anisotropic growth observed at the fiber termini. (page 9, highlighted with green)

Original Figure 6:

Figure 6 | Supramolecular structure and conformational isomerism of UC13. **a**, Schematic representation of directional self-assembly of **UC13**. **b**, Conformational isomerism of **UC13**.

Revised Figure 6:

Figure 6 | Supramolecular structure and conformational isomerism of UC13. **a**, Schematic representation of directional self-assembly of UC13. **b**, Conformational equilibrium of UC13. (page 32, highlighted with green)

Specific Comment 2

In page 14, it is written that These analyses indicate that aggregate formation proceeds with the octamer and tetramer as the nuclei. For me, it was not very clear whether the authors consider that there are two possible nuclei size (octamer and tetramer) or that the nuclei size has size distribution between the octamer and tetramer.

Response to Specific Comment 2

Thank you for your comment. We revised the sentence based on your comments.

Original sentence: These analyses indicate that aggregate formation proceeds with the octamer and tetramer as the nuclei.

Revised sentence: These analyses indicate that aggregate formation involves (at least) two steps, one proceeding through octamer nuclei formation and another through tetramer nuclei formation. (page 14, highlighted with green)

Specific Comment 3

It would be helpful if the authors define *tnuc* and *tvis* in Figure 10.

Response to Specific Comment 3

Thank you for your useful suggestion. We defined *tnuc* and *tvis* in Figure 10.

Original Figure 10:

Figure 10 | Schematics of the two proposed mechanisms for gel formation.

Revised Figure 10:

Figure 10 | Schematics of the two proposed mechanisms for gel formation. (page 37, highlighted with green)

Response to the Comments of Reviewer #3

General Comment

Response to General Comment

Thank you very much.

Response to the Comments of Reviewer #4

(Revisions to the main text and Supplementary Information are highlighted blue)

General Comment

A delightful manuscript to read. The authors have meticulously described and investigated the gelation phenomena of a urea-based gelator relating to describing its self-assembly driven through nucleation processes. The use of HS-AFM has provided some fantastic insights into the numerous steps such gelators can go through. I can only but recommend the publication of the manuscript. I give some small recommendations to try and help improve some aspects of the paper.

The use of crystal structures and their analysis to infer structure motifs/synthons of the supramolecular fibres is a well-taken path within the supramolecular gel community but/and is fraught with mistakes and misunderstandings. The authors have covered this very well and used the crystal data appropriately. Two related things I felt the authors did not quite cover are:

Response to General Comment

We sincerely appreciate your positive feedback and insightful suggestions.

Specific Comment 1

the literature (evidence) of the urea tape hydrogen bonding isn't significant enough to "nail" home the proof of urea assembly and I would recommend adding more discussion and citation of work covering urea assembly (a good place to start would be to cover works by Custelcean, Diaz, Steed, as examples).

Response to Specific Comment 1

Thank you for your important comments. We added references by Custelcean, Diaz, and Steed et al. to the discussion on urea assembly, as related to Specific Comment 2 (newly added ref. 50–53, highlighted with blue). Instead, we deleted the reference on polymer gels in the introduction section, in accordance with the rules regarding the number of references.

Original sentence: The assumption that the self-assembly of **UC13** is driven by hydrogen bonding of ureido groups is based on single-crystal X-ray crystallography of the cyclohexyl analog, **UCCy** (Figure 2d and Supplementary Fig. 5). **UCCy** showed a 1D repeating structure consisting of two-molecules unit with a 1D orientation, where the adjacent chains were arranged antiparallel to each other. The distance between the N–H and C=O moieties of the ureido groups of each **UCCy** (ca. 2.7 Å) indicated that the intermolecular hydrogen bonding of these groups was the major driving force for self-assembly.

Revised sentence: The assumption that the self-assembly of **UC13** is driven by hydrogen bonding of ureido groups is based on single-crystal X-ray crystallography of the cyclohexyl analog, **UCCy** (Figure 2d and Supplementary Fig. 6, See also Supplementary Fig. 7 for information on the X-ray scattering pattern of **UC13**). **UCCy** showed a 1D repeating structure consisting of two-molecules unit with a 1D orientation.^{50–53} The distance between the N–H and C=O moieties of the ureido groups of each **UCCy** (ca. 2.7 Å) indicated that the intermolecular hydrogen bonding of these groups was the major driving force for self-assembly. (page 6, 7, highlighted with blue)

Newly added references

50. Hanabusa, K., Shimura, K., Hirose, K., Kimura, M. & Shirai, H. Formation of Organogels by Intermolecular Hydrogen Bonding between Ureylene Segment. *Chem. Lett.* **25**, 885–886 (1996).
51. Custelcean, R. Crystal engineering with urea and thiourea hydrogen-bonding groups. *Chem. Commun.* 295–307 (2008).
52. Steed, J. W. Anion-tuned supramolecular gels: a natural evolution from urea supramolecular chemistry. *Chem. Soc. Rev.* **39**, 3686–3699 (2010).
53. Häring, M. & Díaz, D. D. Supramolecular metallogels with bulk self-healing properties prepared by in situ metal complexation. *Chem. Commun.* **52**, 13068–13081 (2016). (page 23, highlighted with blue)

Deleted references

Ito, K. Novel cross-linking concept of polymer network: Synthesis, structure, and properties of slide-ring gels with freely movable junctions. *Polym. J.* **39**, 489–499 (2007).

Sun, J. Y. *et al.* Highly stretchable and tough hydrogels. *Nature* **489**, 133–136 (2012).

Sakai, T. Gelation mechanism and mechanical properties of Tetra-PEG gel. *React. Funct. Polym.* **73**, 898–903 (2013).

Nakahata, M., Takashima, Y., Yamaguchi, H. & Harada, A. Redox-responsive self-healing materials formed from host-guest polymers. *Nat. Commun.* **2**, 511 (2011).

Phadke, A. *et al.* Rapid self-healing hydrogels. *Proc. Natl. Acad. Sci. U. S. A.* **109**, 4383–4388 (2012).

Kloxin, A. M., Kasko, A. M., Salinas, C. N. & Anseth, K. S. Photodegradable hydrogels for dynamic tuning of physical and chemical properties. *Science* **324**, 59–63 (2009).

Breger, J. C. *et al.* Self-folding thermo-magnetically responsive soft microgrippers. *ACS Appl. Mater. Interfaces* **7**, 3398–3405 (2015).

Gladman, A. S., Matsumoto, E. A., Nuzzo, R. G., Mahadevan, L. & Lewis, J. A. Biomimetic 4D printing. *Nat. Mater.* **15**, 413–418 (2016).

Specific Comment 2

Observing such large and regular structures from a small molecule's assembly has to make the reader go, how is this different to crystallisation and are they crystalline? The history of crystal growth and the literature of it is significantly vast and when reading such papers it is sometimes noted that the field of supramolecular polymers is reinventing the wheel, somewhat harshly stating so but is factually correct. Invoking periodic assembly in their models' descriptions obviously means the comparison in terms of the physical laws is likely to be needed and justified (how many times have we seen the mathematics repeated when describing similar phenomena from different aspects of science, nucleation and autocatalysis as examples). So with those discussions, I would like to know if the authors have some other evidence of the structural nature of their structures? Have they done any scattering (PXRD, SAXS,

SANS, WAXS etc) to observe any periodic structure and/or secondary evidence of the sizes of their assemblies in situ (Adams and Terech/Weiss have some excellent literature on this)? Are CD, IR, Raman also appropriate tools to find potential correlations between the conformations and the observed structures?

Response to Specific Comment 2

We appreciate your important comments. Although we had estimated the supramolecular structure of **UC13** based on the single-crystal X-ray structure of **UCCy**, we measured SAXS and WAXS at SPring-8 based on your comments (newly added Supplementary Fig. 7). The SAXS patterns of the DMSO- and EMI-Tf₂N gels showed peaks at 1.9 nm^{-1} , corresponding to 32.2 \AA , which is close to the molecular length of **UC13** (2.7 nm) (newly added Supplementary Fig. 7a). This new experimental result suggests the possibility of the formation of a parallel dimer (Revised Figure 6a). Furthermore, the WAXS pattern of the xerogel of **UC13** showed the peak indicating a periodic structure with a spacing of approximately 4 \AA , which was thought to be derived from hydrogen-bonding (newly added Supplementary Fig. 7b). The WAXS patterns of the DMSO- and EMI-Tf₂N gels of **UC13** showed broad peaks (halo) around $2\theta = 10^\circ$, which can be attributed to the periodic structure based on hydrogen-bonding.

Supplementary Figure 7. X-ray scattering pattern of **UC13**. **a**, SAXS patterns of **UC13**-Xerogel (black), **UC13**-DMSO gel (66.5 mM, blue), and **UC13**-EMI-Tf₂N gel (50.0 mM, red). **b**, Enlarged WAXS patterns of **UC13**-Xerogel (black), **UC13**-DMSO gel (66.5 mM, blue), and **UC13**-EMI-Tf₂N gel (50.0 mM, red). A peak at 109.3 Å, corresponding to the 0.6 nm⁻¹ in Supplementary Fig. 7a, was also observed in the blank, in which only the glass capillary was measured. The distance of 32.2 Å, corresponding to the 1.9 nm⁻¹ in Supplementary Fig. 7a, is slightly larger than the molecular length of **UC13** (2.7 nm). For instance, this distance may correspond to a self-assembly in a parallel manner (Figure 6a). Source data of the graphs are provided as a Source Data file. (page S11, highlighted with blue, The gray highlights mean revisions in accordance with the formatting instructions)

In addition, in IR measurements using a lab equipment, we were unable to detect the trace amount of **UC13**-derived peaks because the solvent peak was too strong at the typical gelation concentration.

Based on the above results, the main text was revised as shown in **Response to Specific Comment 1**, and the Supplementary Fig. 7 and Supplementary Methods of XRD Analysis was added.

Newly added Supplementary Methods (XRD Analysis of **UC13**)

XRD Analysis: The gel samples for SAXS analysis were prepared in a quartz glass capillary tube (2.0 mmf) and measured at BL19B2 beamline with a q-range from 0.047 to 33.121 nm⁻¹ under the beamline standard conditions¹⁶ as shown in Supplementary Fig. 7a. The WAXS measurement of gel samples were performed at BL02B2 beamline in SPring-8 with a 2θ range from 1.0 to 83.0 deg using a Lindeman glass capillary tube (0.2 mm) under the beamline standard condition¹⁷ as shown in Supplementary Fig. 7b. (page S5, highlighted with blue)

Newly added references in Supplementary Methods

16. http://www.spring8.or.jp/wkg/BL19B2/instrument/lang-en/INS-0000000300/instrument_summary_view

17. S. Kawaguchi., M. *et al.* *Rev. Sci. Instrum.* **88**, 085111 (2017).

(page S23, highlighted with blue)

Additional Comment

I feel more expansion of the statement is needed for Line 364-365, “, as the existing fiber can catalyse it”. I have had a particular interest in the concept of supramolecular polymers self-catalysing (induction) new polymers/fibres/fibrils. As this is closely linked to the concepts of nucleation and growth dynamics. I have yet to find a good description (based on theoretical or analytical methods) of why we all observe this “catalysis” of fibres. Especially given observations varying from “doing nothing (only convection and/or Brownian motion acting mechanically on structures?)” to sonication to “autocatalysis”. So I would recommend the authors add some content to this statement with some relevant literature to discuss it in the context of their models and observations.

Response to Additional Comment

Thank you for your comment. In the original sentence, we used the word “catalyzes”, but we used it with the meaning of “inducing the nucleus”, rather than implying a meaning such as “autocatalyst”. To clarify our intent, we revised the main text based on the keywords of “nucleation elongation mechanism” and “secondary nucleation”.

Original sentence: As described above, we identified the specific exponents based on the observations on the concentration dependences of three different time scales: the start of fiber growth via nucleation (t_{nuc}), fiber growth into a visible size (t_{vis}), and the appearance of third-step thick fibers in the HS-AFM experiments. These values allowed us to consider two possible mechanisms. The first mechanism, depicted in Figure 10 (path a), assumes that gel formation is limited by the formation rate of the octamer nucleus, and subsequent fiber growth and/or new fiber formation occurs on the surface of existing fibers. The latter may also follow the cooperative mechanism, albeit with a smaller nucleus size of 4 (or 3–5), as the existing fiber can catalyze it. Fibers can also grow on the surface of probe or HOPG-substrate in HS-AFM experiments. These surface reactions may resemble those on existing fibers and proceed via the tetramer nuclei. Thus, the octamer nucleation step can be skipped under these conditions. This explains the fourth-power dependence of the fiber appearance time in HS-AFM images and reason for the fibers appearing faster than the macroscopically observed nucleation time.

Revised sentence: As described above, we identified the specific exponents based on the observations on the concentration dependences of three different time scales: the start of fiber growth via nucleation (t_{nuc}), fiber growth into a visible size (t_{vis}), and the appearance of third-step thick fibers in the HS-AFM experiments. These values allowed us to consider two possible mechanisms. The first mechanism (Figure 10, path a) assumes that gel formation is initiated by the formation of an octamer nucleus, followed by growth of the nucleus to form a short fiber. The resulting fiber may act as a template and recruit the nucleus with a smaller nucleus size of 4 (or 3–5).^{19–23} Fibers can also grow on the surface of probe or HOPG-substrate in HS-AFM experiments. These surface reactions may resemble those on existing fibers and proceed via the tetramer nuclei. Thus, the octamer nucleation step can be skipped under these conditions. This explains the fourth-power dependence of the fiber appearance time in HS-AFM images and reason for the fibers appearing faster than the macroscopically observed nucleation time. In addition, the existing fibers may

cause not only the growth of fibers on their termini, but also the formation of new fibers on the surface through secondary interactions.^{54–58} (page 15, highlighted with blue)

Original sentence: The overall mechanism of supramolecular gel formation of **UC13** is as follows (Figure 1b): **UC13** forms an octamer nucleus, which catalyzes the formation of bundled fibers by accumulating tetramer nuclei on the existing fiber. The fibers formed from the nuclei grow in an intermittent manner repeating the elongation and pause phases, which could be explained by our block stacking model. To conclude, a similar approach to the present study could be applied to other versatile supramolecular gels, especially those that have already proven useful, allowing us to obtain insight into the molecular level mechanism of their formation. This would more spatiotemporal control of gel formation, which would greatly expand the range of gel applications.

Revised sentence: The overall mechanism of supramolecular gel formation of **UC13** is as follows (Figure 1b): **UC13** forms an octamer nucleus, which leads to the formation of bundled fibers by accumulating tetramer nuclei on the existing fiber. The fibers formed from these nuclei grow in an intermittent manner repeating the elongation and pause phases, which could be explained by our block stacking model. These processes follow the cooperative mechanism.^{68–70} To conclude, a similar approach to the present study could be applied to other versatile supramolecular gels, especially those that have already proven useful, allowing us to obtain insights into the molecular level mechanism of their formation. This would lead to more spatiotemporal control of gel formation, which would greatly expand the range of gel applications. (page 16, highlighted with blue)

Newly added references

68. Zhao, D. & Moore, J. S. Nucleation–elongation: a mechanism for cooperative supramolecular polymerization. *Org. Biomol. Chem.* **1**, 3471–3491 (2003).
69. Jonkheijm, P., Schoot, P., Schenning, A. P. H. J., Meijer, E. W. Probing the solvent-assisted nucleation pathway in chemical self-assembly. *Science* **313**, 80–83 (2006).
70. Smulders, M. M. J. *et al.* How to distinguish isodesmic from cooperative supramolecular polymerisation. *Chem. Eur. J.* **16**, 362–367 (2010).

Response to the Comments of Reviewer #1

Reviewers' Comments

The authors have satisfactorily addressed the concerns raised by me. Therefore, I recommend its publication in the current form.

Furthermore, as requested by the editor, I have also looked over the responses of the authors to the concerns from Reviewer #4 as well as evaluated their responses. They seem to be satisfactory enough to be accepted for publication.

Response to Comment

We sincerely appreciate your positive evaluation and recommendation for the publication of our manuscript.

Response to the Comments of Reviewer #2

Reviewers' Comments

The authors have addressed properly the comments from the reviewers. I think that the manuscript has been improved to read clearly. This study provides new insights into supramolecular gelation, and I recommend it to be published in Nature Communications in the revised form.

Response to General Comment

We are grateful for your recommendation for publication. We are pleased that the manuscript has been improved and that you found it valuable in providing new insights into supramolecular gelation.

Response to the Comments of Reviewer #3

Reviewers' Comments

Response to General Comment

Thank you very much.